# ETS transcription factors induce a unique UV damage signature that drives recurrent mutagenesis in melanoma

Peng Mao[1], Alexander J. Brown[1], Shingo Esaki[2], Svetlana Lockwood [3], Gregory M.K. Poon [2,4], Michael J. Smerdon[1], Steven A. Roberts[1,5] & John J. Wyrick [1,5]

Recurrent mutations are frequently associated with transcription factor (TF) binding sites (TFBS) in melanoma, but the mechanism driving mutagenesis at TFBS is unclear. Here, we use a method called CPD-seq to map the distribution of UV-induced cyclobutane pyrimidine dimers (CPDs) across the human genome at single nucleotide resolution. Our results indicate that CPD lesions are elevated at active TFBS, an effect that is primarily due to E26 transformation-specific (ETS) TFs. We show that ETS TFs induce a unique signature of CPD hotspots that are highly correlated with recurrent mutations in melanomas, despite high repair activity at these sites. ETS1 protein renders its DNA binding targets extremely susceptible to UV damage in vitro, due to binding-induced perturbations in the DNA structure that favor CPD formation. These findings define a mechanism responsible for recurrent mutations in melanoma and reveal that DNA binding by ETS TFs is inherently mutagenic in UV-exposed cells.

---

[1] School of Molecular Biosciences, Washington State University, Pullman, WA 99164, USA. [2] Department of Chemistry, Georgia State University, Atlanta, GA 30303, USA. [3] Paul G. Allen School for Global Animal Health, Washington State University, Pullman, WA 99164, USA. [4] Center for Diagnostics and Therapeutics, Georgia State University, Atlanta, GA 30303, USA. [5] Center for Reproductive Biology, Washington State University, Pullman, WA 99164, USA. Correspondence and requests for materials should be addressed to S.A.R. (email: sroberts@vetmed.wsu.edu) or to J.J.W. (email: jwyrick@vetmed.wsu.edu)

U V light induces the formation of cyclobutane pyrimidine dimers (CPDs) and, to a lesser extent, pyrimidine (6-4) pyrimidone photoproducts (6-4PPs) at dipyrimidine sequences in DNA[1]. If unrepaired, these lesions can induce carcinogenic mutations that drive the development of skin cancers like melanoma. Whole-genome sequencing of melanomas has revealed that UV-induced mutation densities are highly variable throughout the genome[2–6]. While variations in mutation density correlate with replication timing, transcription, and nucleotide excision repair (NER) activity, the detailed molecular mechanisms that shape the genomic landscape of UV-induced mutation density are unclear. Elucidating these mechanisms is important, since they contribute to the etiology of recurrent driver mutations in human skin cancers.

While most studies have focused on the origin of recurrent mutations in gene coding regions, many recurrent mutations in cancer genomes are associated with noncoding regulatory elements, such as transcription factor binding sites (TFBS), particularly in melanoma (e.g.,[7,8]). Recent analysis of sequenced melanoma genomes has suggested that mutation densities are elevated not only at specific binding sites (e.g., in the human telomerase (*TERT*) promoter[9,10]), but are generally higher at most TFBS[11]. The underlying cause of elevated mutation densities at TFBS is currently unknown, but possible mechanisms may include reduced displacement and removal of nascent DNA synthesized by DNA polymerase α during replication[12] and reduced repair activity, presumably due to transcription factors (TFs) restricting access of the NER machinery to UV lesions at TFBS[11].

Alternatively, TF binding may alter the rate at which UV lesions form in DNA[13]. An increased frequency of UV damage at TFBS could contribute to elevated mutation rates; however, most TFs appear to suppress UV damage formation at their binding sites[13], likely by restricting the conformational flexibility of DNA needed to form CPD lesions. Our recent survey of UV damage formation across the yeast genome indicated that the well-studied yeast TFs Reb1 and Abf1 primarily suppress the formation of CPD lesions[14]. Less is known about how human TFs affect UV damage levels, due to the relatively low resolution of most published surveys of UV damage in the human genome[6,15,16]. A recent single-nucleotide resolution map of UV damage detected variations in UV damage at TFBS[17] and elevated damage levels at binding sites of a Nuclear Transcription Factor Y subunit (NFYB). However, NFYB binding sites have not been associated with recurrent mutations in melanomas or other cancers (e.-g.,[7,18]), so the extent to which variations in UV damage formation at TFBS contribute to recurrent mutagenesis in human cancers is unknown.

A potential limitation with the few published human UV damage maps (including our own study[15]) is that all employed an immunoprecipitation step using a CPD-specific antibody, which in some cases have been reported to have sequence specificity[19]. For example, lesions associated with TT (and CT) dipyrimidines may be over-represented relative to those occurring in other sequence contexts when immunoprecipitated with certain antibodies (e.g., TDM-2[6,15,19]). TT dimers, however, are typically not mutagenic, due to error-free bypass by DNA polymerase η[20,21]. Instead, most UV-induced mutations are C > T substitutions in TC and CC dipyrimidine sequence contexts[18,22]. To determine how differences in the rate of UV damage formation influence mutagenesis at TFBS, it is important to accurately measure CPD lesions occurring in all dipyrimidine sequence contexts.

Here, we describe a single-nucleotide resolution map of CPD lesions in UV-irradiated human cells, produced by a method known as CPD-seq[14] that does not require an immunoprecipitation step to map UV damage (Fig. 1a). Our CPD-seq data

indicate that CPD lesions are generally elevated at active TFBS in UV-irradiated human skin cells, an effect that is primarily due to oncogenic E26 transformation-specific (ETS) TFs. We show that ETS TFs induce a unique signature of UV damage, both in cells and in vitro, which is highly correlated with recurrent mutations in melanoma. Repair activity is high at ETS binding sites, indicating that increased damage formation is likely the primary mechanism driving elevated mutation rates at these sites.

## Results

**TFBS have elevated levels of UV-induced damage and mutations.** To test the hypothesis that variations in UV damage formation contribute to elevated mutation rates at TFBS, we analyzed mutation densities and UV damage levels at a well-defined set of TFBS for 82 distinct human TFs[23], initially focusing on TFBS located in promoter regions. We analyzed ~21 million somatic mutations in melanoma genomes from 184 donors sequenced by the International Cancer Genome Consortium[18]. Mutagenic bypass of UV-induced lesions produced the vast majority of these mutations, as indicated by ~97% of the mutations (primarily C-to-T substitutions) occurring in dipyrimidine sequences (Supplementary Fig. 1a). Using this much larger ICGC melanoma dataset[18], we confirmed that mutation density is significantly elevated near active TFBS (Fig. 1b), defined as TFBS that overlap with DNase I hypersensitive sites (DHS) in human melanocytes[11]. In contrast, there is only a minor difference in mutation density at inactive TFBS (Fig. 1c), indicating that mutation density is specifically elevated at active TFBS.

To measure the levels of UV damage at TFBS, we employed our recently developed CPD-seq (cyclobutane pyrimidine dimer-sequencing) method[14] to map the genome-wide distribution of CPD lesions at single nucleotide resolution in normal human fibroblasts (NHF1) harvested immediately following UV irradiation. Our human CPD-seq map of UV-irradiated NHF1 cells (100 J m$^{-2}$ of UV-C light) displayed a significant enrichment of reads associated with lesions at dipyrimidine sequences (i.e., TT, TC, CT, and CC; see Supplementary Fig. 1b). CPD-seq reads associated with dipyrimidine sequences were also significantly enriched in UV-irradiated NHF1 cells relative to the matched unirradiated control (~10-fold in Supplementary Fig. 1b). The relative abundances of TT, TC, CT, and CC sequences correspond well with both the known reactivity of these sequences following UV-irradiation and our published CPD map for UV-irradiated yeast cells[14]. Additionally, the CPD-seq data generally lacked the strong strand bias among highly transcribed genes (Supplementary Fig. 1c, d) that is typically generated by transcription-coupled NER, indicating our data comprises an accurate map of initial CPD lesion formation. We sequenced CPD-seq libraries from three independent experiments, which in total comprise 112 million sequencing reads.

Our CPD-seq data revealed that CPD lesions are significantly elevated (~1.7-fold) near the center of active TFBS relative to flanking DNA (Fig. 1d). This effect is very reproducible across multiple biological replicates (Supplementary Fig. 2a), including cells treated with a lower, more physiologically relevant dose of UV light (20 J m$^{-2}$, see Supplementary Fig. 2b). However, no enrichment of CPDs is evident at TFBS in UV-irradiated naked DNA (Fig. 1d and Supplementary Fig. 2c), and only a small enrichment in CPD levels is observed at inactive TFBS (Fig. 1e), indicating that DNA binding by TFs is required for elevated UV damage formation. Since UV-induced mutations occur primarily in cytosine-containing CPDs, we also analyzed the levels of mutagenic CPD lesions (mCPDs), corresponding to lesions at TC, CT, and CC dipyrimidines. Our data indicate that mCPD levels are also elevated at active TFBS (Supplementary Fig. 3). Taken

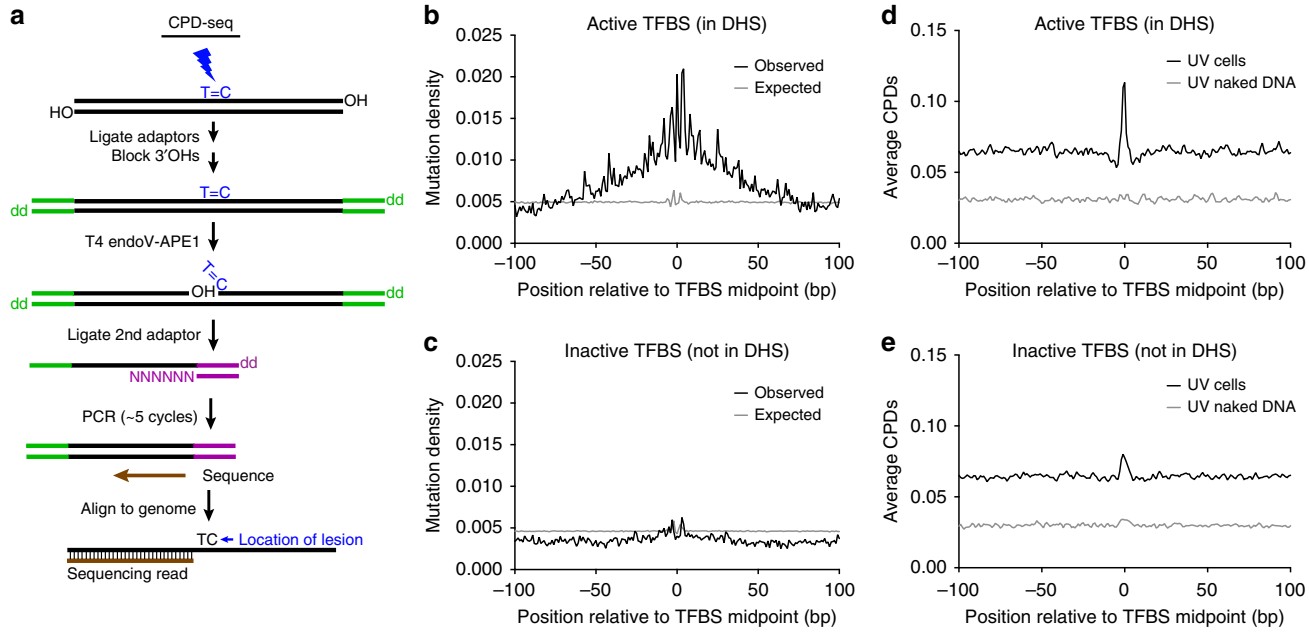

**Fig. 1** Genome-wide map of CPD lesions reveals that CPDs are elevated at active TFBS. **a** Schematic diagram of the CPD-seq method for mapping CPD lesions at single nucleotide resolution. 'T = C' indicates a CPD lesion at TC dipyrimidine. Oligonucleotide adapters are indicated in green and purple; 'NNNNNN' indicates a random DNA hexamer. A 3′ hydroxyl is indicated with OH, while 'dd' indicates a dideoxy 3′ end. The CPD lesion is cleaved with T4 endonuclease V and apurinic/apyrimidinic endonuclease (APE1) to generate a free 3′ hydroxyl immediately upstream of the CPD lesion, which is ligated to an adapter and sequenced. **b** Mutation density surrounding active promoter-proximal TFBS from 184 sequenced melanoma tumors[18]. Observed mutation density (i.e., in melanoma tumors) was analyzed adjacent to known TFBS located in promoter-proximal regions (up to 2500 bp upstream of transcription start site) that were considered active (i.e., overlapping with melanocyte DNase I-hypersensitivity (DHS) regions) for 82 distinct TFs. Expected mutation density was determined from the corresponding DNA sequences surrounding each active promoter-proximal TFBS, based on the trinucleotide mutation signature frequencies for all promoter-proximal regions. **c** Same as part (**b**), except mutations were analyzed adjacent to promoter-proximal TFBS that were considered inactive (i.e., not overlapping with melanocyte DHS regions). **d** Average number of CPD lesions (per TFBS) adjacent to active promoter-proximal TFBS. CPD lesions were mapped using CPD-seq from UV-irradiated NHF1 cells (100 J m$^{-2}$) or isolated NHF1 DNA that was UV-irradiated (80 J m$^{-2}$) in vitro (naked DNA). Cellular DNA was harvested immediately after UV irradiation, so essentially no repair was allowed to occur. **e** Same as in part (**d**), except CPD lesions were analyzed adjacent to inactive promoter-proximal TFBS

together, these findings indicate CPD lesions form at a higher frequency, on average, at active TFBS in UV-irradiated human cells, which correlates with high mutation frequency. However, the mutation peak at active TFBS is broader than the CPD peak; therefore, we investigated the impact of individual TFs on CPD formation and mutagenesis in more detail.

**Elevated UV damage and mutations are associated with ETS TFBS.** To test whether elevated CPD levels are associated with specific TFs in our dataset, we analyzed CPD enrichment (CPDs in cells/CPDs in naked DNA) for each of the 82 human TFs[23]. This analysis identified five TFs (ELK4, ETS1, GABPA, NFYA, and NFYB) that show significant enrichment of CPD levels in cells and significantly contribute to the overall CPD count at active TFBS (Fig. 2a). These TFs do not affect CPD formation in DNA immediately flanking the TFBS (Fig. 2b), indicating that CPD induction is specifically associated with the actual binding site. Elevated CPD levels have been previously reported for NFYB binding sites[17], consistent with our results; however, elevated CPD levels at ELK4, ETS1, and GABPA binding sites have not been reported previously. Of these five TFs, only ELK4, ETS1, and GABPA significantly contribute to the enrichment of mCPDs (Fig. 2c), as NFYA and NFYB primarily stimulate the formation of thymine dimers (lesions at TT dipyrimidines). While cellular mCPD levels are enriched at core TFBS for ELK4, ETS1, and GABPA, there is no enrichment in mCPDs in DNA flanking these TFBS (Fig. 2d).

ELK4, ETS1, and GABPA are members of the E26 transformation-specific (ETS) family of TFs[24], suggesting that a

common DNA-binding mechanism shared by these three ETS TFs stimulates the formation of CPD lesions. An additional ETS family member (ELF1) does not show the same degree of CPD or mCPD enrichment (Fig. 2a, c), possibly because it is in a different ETS subfamily (class II) than the ELK4, ETS1, GABPA TFs (class I), and thus may have an altered binding mechanism[24]. Other TFs also show a significant increase in mCPD levels, notably RFX5 and NRF1, but these TFs contribute less to the overall number of mCPDs at active TFBS (Fig. 2c).

We also analyzed the enrichment of observed mutations relative to expected (based on DNA sequence context) for each TF. Mutation enrichment was highest for ETS family TFs (Fig. 2e), indicating that elevated mCPD levels correlates with elevated mutation density at ETS binding sites. In general, enrichment of mCPD lesions (relative to naked DNA) correlates with mutation enrichment across all active TFBS (Fig. 2f; two-sided Spearman's $\rho = 0.47$, $P = 0.0002$), indicating a tight coupling between initial damage levels and mutation density at TFBS. Interestingly, while CPD lesions and mutation density are elevated at ETS family TFBS, the binding sites of many other TFs, including members of the c-Fos and c-Jun TF families, are depleted of CPDs (Fig. 2), indicating that TF binding may protect these sites from CPD formation, as has previously been reported for certain yeast transcription factors[14].

**Elevated CPD formation and repair activity at ETS binding sites.** Analysis of the distribution of CPDs around ETS family TFBS (i.e., ELF1, ELK4, ETS1, and GABPA) revealed that both

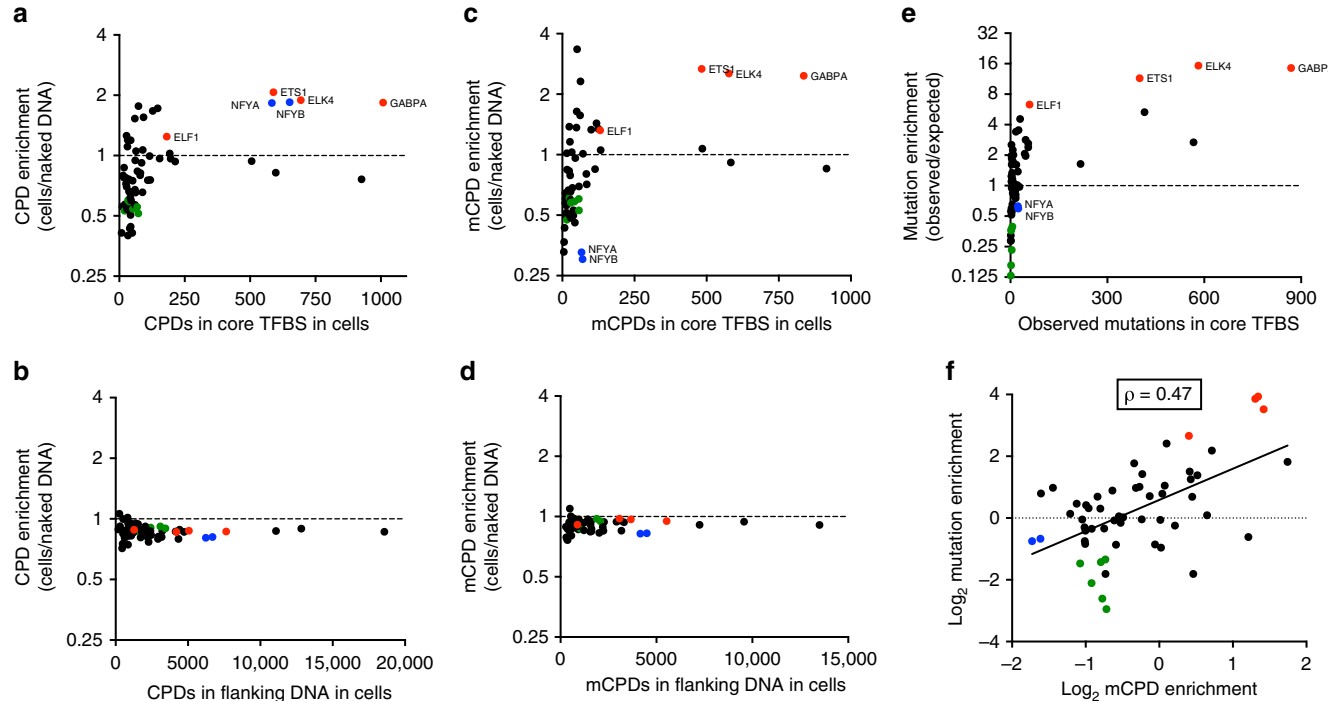

**Fig. 2** Elevated CPD lesions and mutation density are associated with ETS TFBS. **a** Enrichment of CPD levels following UV irradiation in cells relative to naked DNA at active promoter-proximal TFBS for each individual TF. The scaled CPD enrichment was plotted against the total number of CPD lesions present in the set of TFBS for each TF. Only CPD lesions located in the core of the TFBS (−4 to +4 bp relative to the TFBS midpoint) were analyzed. ETS family TFs are in red, NFYA and NFYB are in blue, Fos and Jun TF family members are in green. **b** Same as in part (**a**), except only CPD lesions associated with flanking DNA (−100 to −5 bp and +5 to +100 bp relative to the TFBS midpoint) were analyzed. **c** Same as in part (**a**), except only mutagenic CPDs (mCPDs), which are CPD lesions at TC, CT, and CC dinucleotides, were analyzed. **d** Same as part (**c**), except only mCPD lesions associated with flanking DNA (−100 to −5 bp and +5 to + 100 bp relative to the TFBS midpoint) were analyzed. **e** Enrichment of melanoma mutations at the core of active promoter-proximal TFBS (−4 to +4 bp relative to the TFBS midpoint) for each TF. Mutation enrichment was calculated from the ratio of observed mutations in 184 melanoma tumors relative to the expected number of mutations based on DNA sequence (see Fig. 1b legend). ETS family TFs are in red, NFYA and NFYB are in blue, and Fos and Jun TF family members are in green. **f** Mutation enrichment strongly correlates with mutagenic CPD (mCPD) enrichment at active TFBS (two-sided Spearman correlation coefficient = 0.47, P = 0.0002). Log–Log plot of mutation enrichment relative to mCPD enrichment for the core of active promoter-proximal TFBS for each TF. Only TFs that passed the inclusion criteria for parts (**c**) and **e** were plotted

CPD and mCPD abundance is significantly elevated (~4-fold and ~5-fold, respectively) near the center of active promoter-proximal ETS binding sites relative to flanking DNA in UV-irradiated cells (Fig. 3a, b). In contrast, the enrichment of CPD and especially mCPD lesions is reduced when ELF1, ELK4, ETS1, and GABPA binding sites are excluded from the set of 82 TFs (Fig. 3d, e), indicating that elevated mCPD formation is almost entirely due to ETS TFs. Similarly, mutation density near the center of ETS binding sites is enriched ~18-fold relative to distal flanking DNA (Fig. 3c). Importantly, this effect is largely independent of DNA sequence context, as the expected mutation density did not show the same trend (Fig. 3c). In contrast, non-ETS TFBS showed a broad but relatively modest increase in mutation enrichment (nearly 10-fold less than for ETS TFBS; Fig. 3f).

A previous study concluded that mutation enrichment at active TFBS is driven by lower repair activity at these sites[11,25], presumably because TF binding inhibits access of the NER machinery to DNA lesions. We examined whether repair activity is inhibited at ETS binding sites by analyzing published XR-seq data for CPD lesions in NHF1 cells[5], which provides a measure of NER activity. Surprisingly, this analysis revealed high levels of CPD repair activity at ETS binding sites (Fig. 3g). This was true not only at the 1 h repair time point, but also at 4 and 8 h following UV irradiation (Supplementary Fig. 4a, b). To confirm this result, we also analyzed repair activity at active ETS TFBS

located outside of promoter regions. Again, we found that repair activity is high at ETS binding sites relative to flanking DNA (Fig. 3h). In contrast, non-ETS binding sites are associated with broadly lower levels of CPD repair activity (Fig. 3i), consistent with the previous study[11]. These data suggest that TFs can stimulate UV mutagenesis through two distinct mechanisms: (1) broad inhibition of repair (most TFs), which yields a relatively small increase in mutation density; and (2) specific induction of UV damage (primarily ETS TF family), which causes a massive increase in mutation density.

**ETS TFs induce unique UV damage and mutation signatures.** To characterize the pattern of UV damage at ETS binding sites, we analyzed our CPD-seq data at high resolution among aligned versions of the different ETS binding site motifs for ELK4, ETS1, and GABPA. This analysis revealed a unique signature of CPD lesions associated with active ETS binding sites. There is a major CPD peak at position −1/0 (i.e., CTTCCGG, underline indicates lesion-forming dipyrimidine) and position 0/+1 (CTTCCGG) relative to the ETS motif midpoint (see black line in Fig. 4a). CPD lesions at these positions are significantly enriched relative to the naked DNA control (scaled by total counts of lesions in promoter regions, to account for differences in sequencing depth), indicating that this CPD peak is not simply a consequence of DNA sequence biases in the ETS motif, but is induced by ETS TF binding.

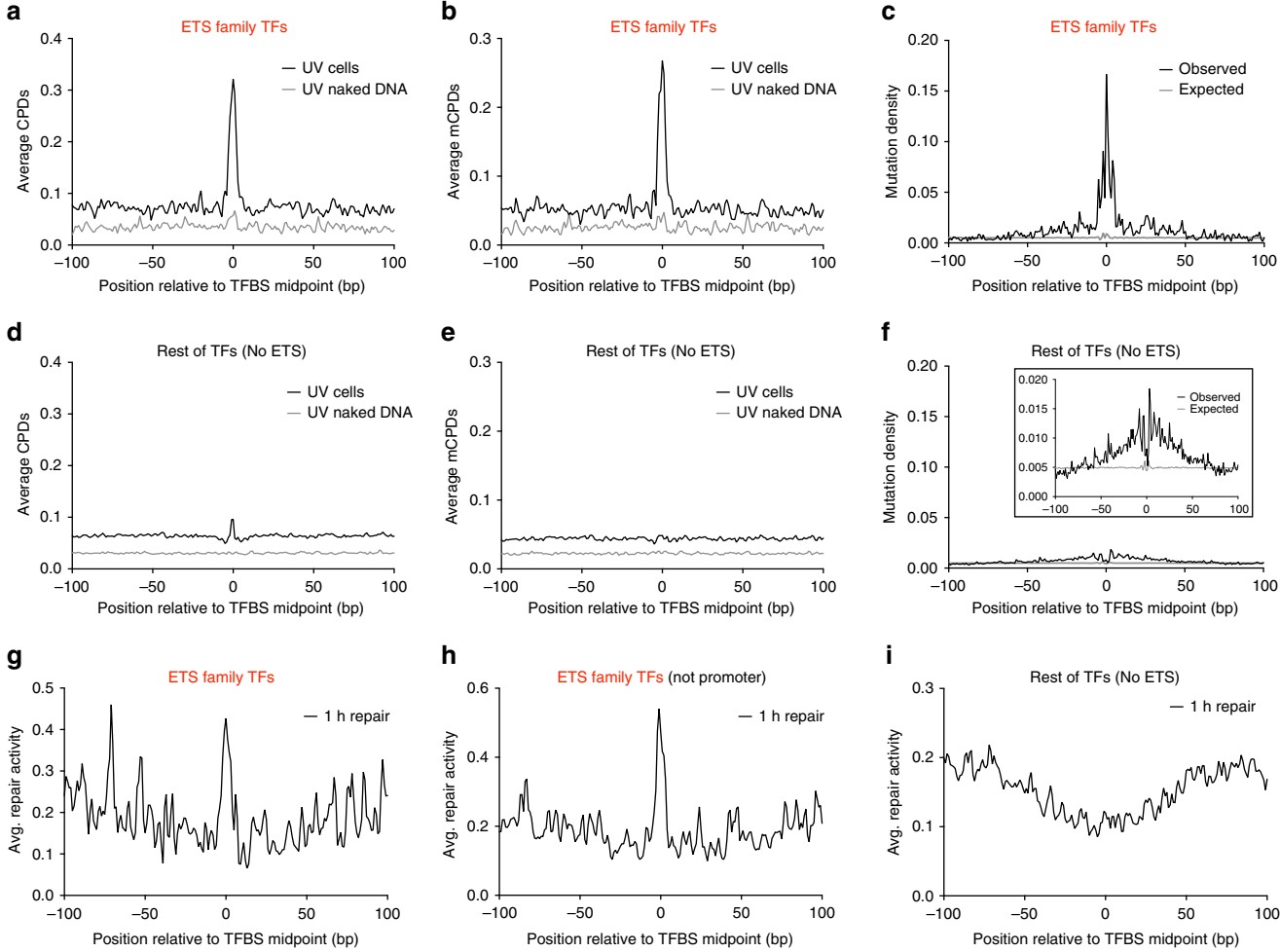

**Fig. 3** ETS family TFBS are the primary contributors to elevated CPD levels. **a, b** Formation of CPD (**a**) and mCPD (**b**) lesions is significantly stimulated at active, promoter-proximal ETS family TFBS (i.e., ELF1, ELK4, ETS1, and GABPA) in UV-irradiated NHF1 cells, but not when isolated NHF1 DNA was UV-irradiated in vitro (naked DNA). **c** Mutation density is significantly elevated at active promoter-proximal ETS family TFBS (i.e., ELF1, ELK4, ETS1, and GABPA) in melanoma tumors, correlating with the higher initial mCPD lesion density at these sites. **d, e** Formation of CPD (**d**) and mCPD (**e**) lesions is not significantly stimulated at active promoter-proximal TFBS after excluding ELF1, ELK4, ETS1, and GABPA binding sites. **f** Mutation density is only slightly elevated surrounding non-ETS family TFBS in aggregate (see inset with expanded scale). Only active promoter-proximal TFBS were analyzed. **g** CPD repair activity is elevated at ETS family TFBS following UV irradiation of human cells. Average CPD repair activity at 1 h repair in UV-irradiated NHF1 cells at ETS family TFBS (i.e., ELF1, ELK4, ETS1, and GABPA). CPD repair activity was calculated using the average number of XR-seq reads[5] at locations surrounding active, promoter-proximal ETS binding sites. XR-seq reads were localized to the putative dipyrimidine lesion associated with each sequencing read. **h** Same as in **g**, except repair activity for active ETS family TFBS located outside promoter regions was analyzed. **i** Same as (**g**), except CPD repair activity was analyzed at non-ETS family TFBS

We observed a second CPD peak at positions −4/−3 relative to the ETS motif midpoint (Fig. 4a). Although this CPD peak is less prominent, it is highly enriched relative to the scaled naked DNA control (~16-fold increase). Positions −4/−3 in the ETS motif consist primarily of purine-pyrimidine dinucleotides (Fig. 4a), which are typically not CPD forming. Only 12% (156 sites) contain a dipyrimidine on the consensus strand at positions −4/−3. Analysis of just these 156 sites revealed the high degree of CPD lesion enrichment at positions −4 and −3 in the ETS motif, relative to the naked DNA control (Fig. 4b). This analysis indicates that ETS binding sites contain a unique UV damage signature consisting of CPD hotspots at positions −4/−3, −1/0, and 0/+1 relative to the motif midpoint.

We next tested whether this UV damage signature correlates with elevated mutation rates. Analysis of the ICGC mutation data for melanomas revealed multiple peaks of mutation density within the aligned ETS binding sites, which are highly correlated

with the ETS UV damage signature (Fig. 4a). For example, the highest mutation density occurs at a conserved C nucleotide at position 0 (i.e., CTTCCGG, underline indicates the mutation site). This mutation peak correlates with CPD peaks at positions −1/0 (TC) and 0/+1 (CC). Analysis of individual ETS TFBS identified many instances of recurrent mutations in melanoma occurring at position 0 in individual ETS sites (Supplementary Table 1). A similar pattern of UV damage formation and mutagenesis was also observed among active ETS TFBS that were not located within a proximal promoter region (e.g., 5′ UTR, intergenic regions, etc., see Supplementary Fig. 5).

A second peak of mutation density occurs at positions −4 and −3 in the ETS motif (Fig. 4a), which coincides with a CPD peak at this same location. Analysis of the 156 sites containing a dipyrimidine sequence at these positions revealed an even more striking enrichment in mutation density at positions −4 and −3 in the ETS motif, as mutation density was ~120-fold and ~240-

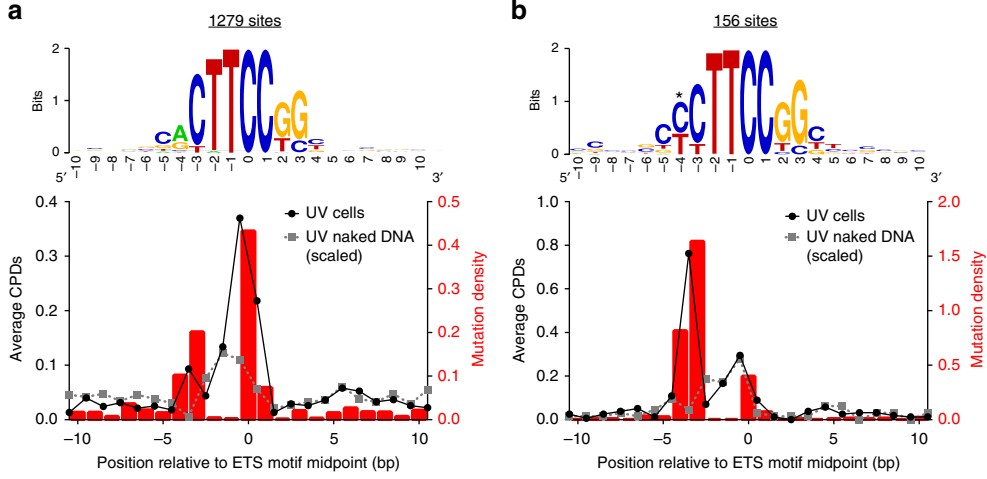

**Fig. 4** CPDs and mutations are elevated at specific locations in ETS binding sites. **a** UV-induced CPD formation and mutation density is enriched at 1279 active, promoter-proximal TFBS for the ETS TFs ELK4, ETS1, and GABPA. TFBS were aligned based on DNA strand and location of ETS consensus sequence. The upper panel depicts the information content of the DNA sequences for the aligned TFBS, matching the known ETS consensus motif. Sequence logo was generated using the weblogo tool[57]. The lower panel plots the mutation density for 184 melanoma tumors relative to average CPD levels following UV-irradiation of NHF1 cells and isolated DNA in vitro (naked DNA). CPD values are plotted at half integer locations, which reflect the average number of CPD lesions forming between the two adjacent nucleotides. CPD lesion density for naked DNA was scaled so that the total CPD levels in promoter-proximal regions in cells and naked DNA were equivalent. **b** Analysis of CPD lesion formation and mutation density at a subset of ETS TFBS (156 sites) that have a pyrimidine nucleotide at position −4 relative to the ETS motif midpoint (indicated with *), and thus can form CPD lesions between position −4/−3. Upper and lower panels are the same as in part (**a**), plotted for this TFBS subset

fold higher than expected at positions −4 and −3, respectively (Fig. 4b). A number of highly recurrent mutations are located at these positions in the ETS motif (Supplementary Table 2), some of which have been previously identified in other melanoma samples[7,8,26]. Our data suggest that a ~16-fold increase in UV damage formation due to ETS binding is likely responsible for these mutation hotspots.

**DNA binding by ETS1 stimulates CPD formation in vitro.** To test whether ETS TF binding to DNA directly induces this unique UV damage signature, we analyzed CPD levels following in vitro UV irradiation of DNA templates containing ETS family TFBS in the presence and absence of purified ETS1 protein. ETS1 protein was chosen for these experiments because its binding sites were associated with elevated CPD formation and mutation density in human cells (Fig. 2), and because it is the most highly expressed ETS family member in primary melanocytes, based on ENCODE RNA-seq data (Supplementary Table 3). As depicted in Fig. 5a, we analyzed UV-induced CPD levels in two model DNA sequences: (1) a highly mutated ETS binding site in the *RPL13A* (Ribosomal Protein L13A) promoter, since this site is the most frequently mutated ETS motif in the ICGC melanoma dataset (see Supplementary Table 1 and 2); and (2) ETS binding sites in the promoter of the *SDHD* (Succinate Dehydrogenase Complex Subunit D) gene (Supplementary Fig. 5a), since mutations at these sites have been linked to decreased *SDHD* expression and poor prognosis in melanoma patients[27].

Purified ETS1 protein (residues 280–441) bound both the *RPL13A* and *SDHD* promoter fragments when the respective radiolabeled DNA is incubated with increasing ETS1 protein in gel shift assays (Fig. 5b, c). The two shifted bands for the *RPL13A* promoter fragment observed at low ETS1 protein concentrations may reflect binding of ETS1 to two ETS motifs in this promoter region (Fig. 5b). Multiple shifted bands were also observed for the *SDHD* promoter fragment (Fig. 5c), consistent with the presence of three ETS binding motifs in this DNA fragment. Importantly, we confirmed that the supershifted bands in the *SDHD* promoter corresponded to sequence-specific interactions between ETS1 and

the ETS motifs, as engineered point mutations in either ETS motif-1 or ETS motif-2 significantly alter the gel shift pattern (Supplementary Fig. 6).

We characterized the impact of ETS1 binding on the frequency of CPD formation by UV irradiating the *RPL13A* and *SDHD* promoter fragments in vitro in the presence or absence of DNA-bound ETS1 protein. The resulting CPD lesions were converted to single strand breaks by specific digestion with T4 endonuclease V, and subsequently detected on sequencing gels. Our data indicate that ETS1 binding to the *RPL13A* promoter fragment significantly induces CPD levels upon UV irradiation at specific locations in the canonical ETS motif-2. Only low levels of CPDs are detectable after UV irradiation in the absence of ETS1, but CPD levels are significantly elevated at two locations in ETS motif-2 upon ETS1 binding (Fig. 5d, see upper and lower bands in ETS motif-2). Sequencing gel analysis indicated that the lower band, in which CPD levels are induced up to ~40-fold upon ETS1 binding, has a size of ~15.5 nt, based on comparison with synthesized oligonucleotides with the same DNA sequences and known lengths (Supplementary Fig. 7). This DNA fragment size is consistent with CPD formation occurring at a $T_{16}$-$C_{17}$ dinucleotide near ETS motif-2, since T4 endonuclease V cleaves the N-glycosidic bond of the 5′ pyrimidine in the CPD, in addition to generating a strand break between the 5′ and 3′ pyrimidines[1]. The $T_{16}$-$C_{17}$ dinucleotide is located at position −4/−3 relative to ETS motif midpoint, corresponding to the precise location of an ETS-associated CPD hotspot identified by CPD-seq; furthermore, the 3′ deoxycytidine ($C_{17}$) is mutated in 47 out of 184 melanoma samples in the ICGC dataset (Supplementary Table 2). The upper band, in which CPD levels are induced up to 20-fold upon ETS1 binding, has a size consistent with CPD formation occurring at the $T_{19}$-$C_{20}$ dinucleotide within the ETS motif-2 (Supplementary Fig. 7). This corresponds to the CPD hotspot identified by CPD-seq at position −1/0 relative to the ETS motif midpoint. The 3′ deoxycytidine ($C_{20}$) of this dipyrimidine is also frequently mutated in melanomas (10 out of 184 tumors, see Supplementary Table 1). In contrast, CPD levels for the $T_{18}$-$T_{19}$ dinucleotide in the ETS motif (−2/−1

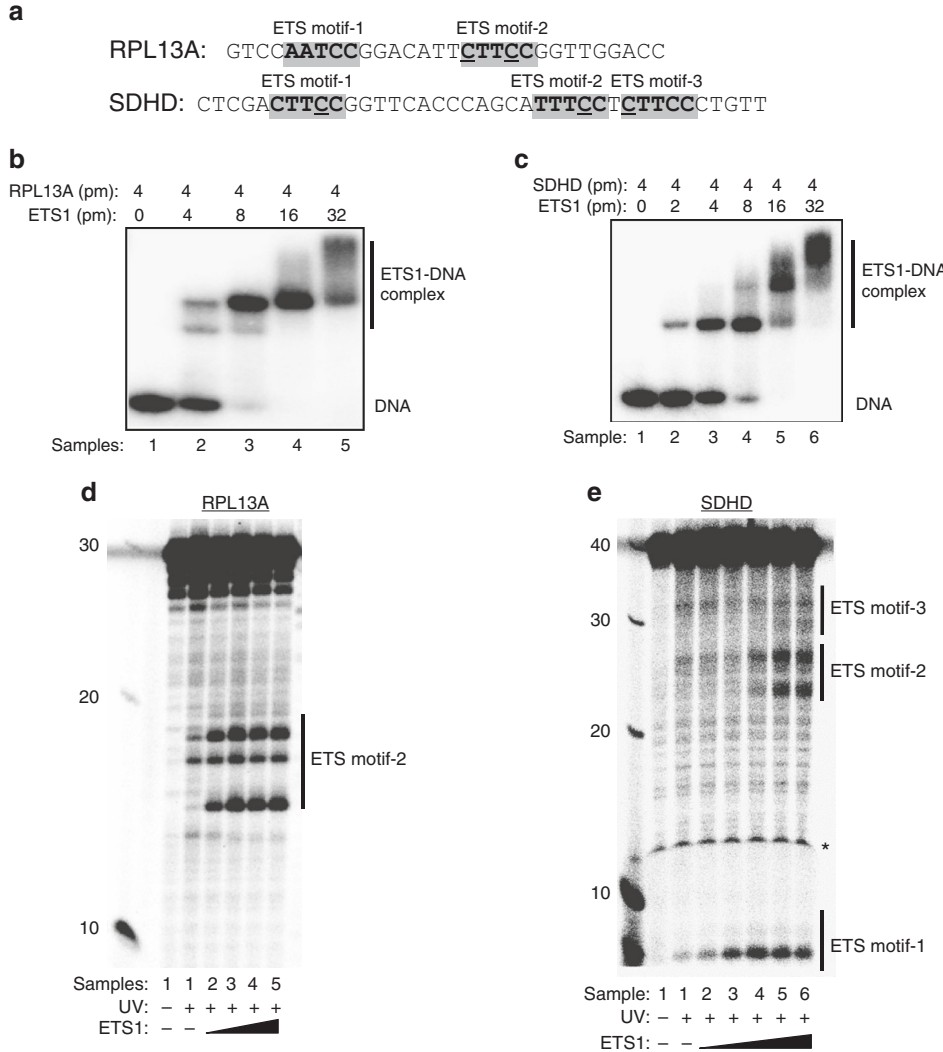

**Fig. 5** Binding of ETS1 protein promotes UV damage formation in vitro. **a** DNA sequences of *RPL13A* and *SDHD* promoter fragments, corresponding to chromosome coordinates chr19:49990710-49990681 and chr11:111957515-111957553, respectively. Putative ETS motifs are shown in gray background and highlighted in bold. Recurrent mutated sites in melanomas are underlined. **b, c** Gel shift assays showing binding of purified ETS1 protein to radiolabeled *RPL13A* (**b**) and *SDHD* (**c**) promoter fragments, respectively. **d** A representative sequencing gel (15%) showing CPD formation in naked *RPL13A* (sample 1, with UV irradiation) and ETS1-bound *RPL13A* DNA (samples 2–5, UV irradiation). The binding products shown in part (**b**) were irradiated with 1KJ m$^{-2}$ of UV-C light and CPD lesions were converted to single strand breaks by T4 endonuclease V digestion. The resulting DNA breaks were separated on a 15% denaturing sequencing gel to analyze damage abundance at different locations. A negative control (naked DNA without UV irradiation) was also digested with T4 endoV to show the background level of DNA cleavage in the absence of UV-induced DNA lesions. The first lane on the left shows a 10-nt DNA ladder. **e** Same as in part (**d**), except the *SDHD* promoter fragment was analyzed on a 12% gel. Asterisk indicates gel running artifact caused by bromophenol blue in the gel loading buffer. Both *RPL13A* and *SDHD* CPD formation experiments were conducted at least 3 times independently with consistent results

relative to the motif midpoint) are not significantly altered by ETS1 binding (Fig. 6d and Supplementary Fig. 7, middle band in ETS motif-2), consistent with our CPD-seq data (Fig. 4).

ETS1 binding also significantly stimulates the formation of CPD lesions at multiple locations in the *SDHD* promoter (Fig. 5e). Even relatively low concentrations of ETS1 protein stimulated CPD formation at the $T_8$-$C_9$ dipyrimidine in the high affinity ETS motif-1 site (Fig. 5e). CPD levels at this location, which is at position −1/0 relative to the motif midpoint, are induced up to 7-fold upon ETS1 binding, consistent with our CPD-seq data. An engineered mutation disrupting the ETS motif-1 (*SDHD*-mt1 in Supplementary Fig. 6a) significantly reduces binding to the *SDHD* promoter fragment at low ETS1 concentrations (Supplementary Fig. 6b, left panel), confirming that this is the high affinity ETS1 binding site in the *SDHD* promoter, and abolishes UV-induced

CPD formation at this site (Supplementary Fig. 6c, left panel, site #1). CPD formation is also induced by higher ETS1 concentrations at two locations in the lower affinity ETS motif-2. Although the precise location of these lesions is more difficult to resolve on the sequencing gels, they likely occur at $T_{24}$-$T_{25}$ and $C_{27}$-$C_{28}$ (Fig. 5e and Supplementary Fig. 6c), corresponding to positions −3/−2 and 0/+1 relative to the motif midpoint, respectively. Position 0/+1 is a CPD hotspot in our CPD-seq data (Fig. 4a). An engineered mutation disrupting ETS motif-2 (*SDHD*-mt2 in Supplementary Fig. 6a) affects the gel shift pattern at higher ETS1 concentrations (Supplementary Fig. 6b, right panel), confirming that this is a low affinity ETS site. Notably, *SDHD*-mt2 abolishes CPD formation not only at the mutated dipyrimidine ($C_{27}$-$C_{28}$), but also the neighboring $T_{24}$-$T_{25}$ dipyrimidine (Supplementary Fig. 6c, right panel, sites #2 and #3), indicating that binding of

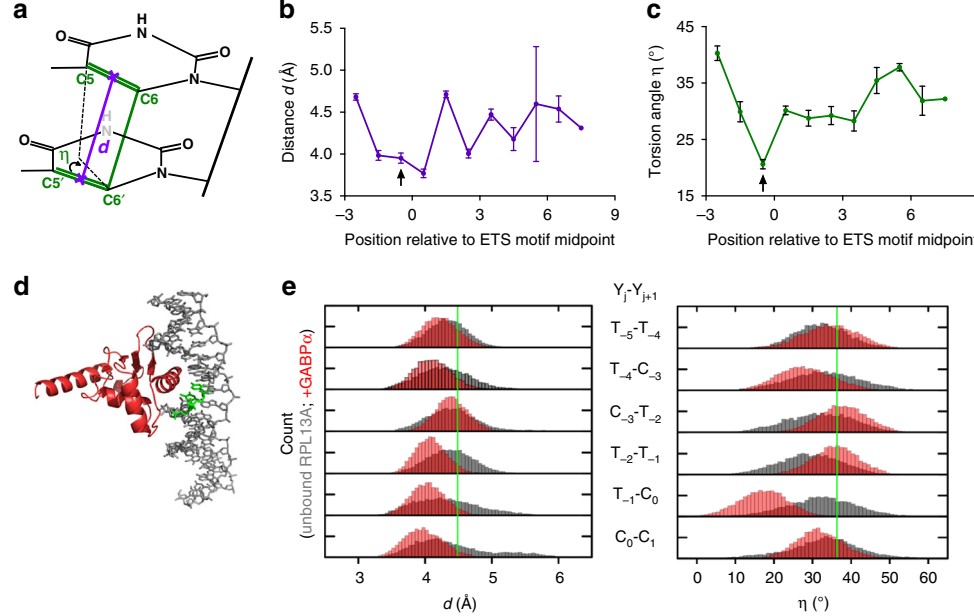

**Fig. 6** ETS binding induces a DNA conformation predisposed to form CPD lesions. **a** Schematic showing the key structural parameters that affect the rate of CPD formation, namely the distance ($d$) between the midpoints of adjacent C5–C6 bonds, and the torsion angle ($\eta$) between the adjacent C5–C6 bonds. **b**, **c** Plots of distance ($d$) and torsion angle ($\eta$) between adjacent C5–C6 bonds for structures of ETS bound DNA. The arrow indicates the location of the CPD hotspot at position −1/0, corresponding to position −0.5 in the plot. The mean ± SEM is plotted for data derived from thirteen crystal structures. **d** Structure of GABPA bound to DNA (PDB ID: 1AWC). The CPD hotspot at positions −1/0 relative to the motif midpoint is highlighted in green. Image was visualized using pymol. **e** Molecular dynamics simulation of GABPA (i.e., GABPα) bound to a 14 nt duplex DNA sequence containing ETS motif-2 in the *RPL13A* promoter. The distance ($d$) and torsion angle ($\eta$) between C5–C6 bonds of adjacent pyrimidines was plotted for the final 100 ns (10,000 frames). The distribution for GABPA-bound DNA is depicted in red, while simulated unbound DNA is shown in gray. Each increment in the ordinate represents 500 counts. Green lines indicate canonical values of B-form DNA

ETS1 to ETS motif-2 is required for damage induction. A number of recurrent *SDHD* promoter mutations in melanoma[27], particularly in ETS motif-1, are associated with these ETS1-induced CPD hotspots (Supplementary Fig. 5 and 6). These in vitro data demonstrate that ETS1 binding causes UV damage hotspots that recapitulate the UV damage signature identified in human cells.

**ETS binding induces a DNA conformation poised to form CPDs**. To investigate the molecular mechanism responsible for the extreme susceptibility of ETS1 binding sites to UV damage, we analyzed published structures of ETS1 bound to DNA. While the biophysics of the UV-induced [2+2] cycloaddition reaction, which joins the C5–C6 double bonds of adjacent pyrimidines to form a CPD lesion, are not fully understood[28], two structural features are believed to regulate the rate of CPD formation: (1) the distance ($d$) between the C5–C6 double bonds in adjacent pyrimidines, and (2) the torsion angle ($\eta$) between the C5–C6 double bonds of adjacent pyrimidines (Fig. 6a). Dipyrimidine sequences in which the C5–C6 bonds are in close proximity ($d <$ 4 Å) and are favorably aligned ($\eta \sim$ 20–30°) generally have a higher quantum yield of UV-induced CPD lesions[28–31].

Analysis of thirteen available ETS1-DNA structures revealed that ETS1 binding modulates the distance ($d$) and torsion angle ($\eta$) between the C5–C6 double bonds of adjacent pyrimidines at specific locations in the ETS motif. The average distance between C5–C6 bonds in these structures is less than 4.0 Å near the midpoint of the ETS motif (Fig. 6b), including the CPD hotspots at positions −1/0 (indicated with an arrow in Fig. 6b) and positions 0/+1, which have average distances of 3.95 ± 0.06 (mean ± SEM, $n = 20$) and 3.77 ± 0.05, respectively. The torsion angle between the C5–C6 double bonds is also particularly

favorable at the TC dipyrimidine at position −1/0 in the ETS motif (indicated with an arrow in Fig. 6c), having an average value of 20.6° ± 0.8° in these structures, which is significantly smaller than those at flanking positions. This trend remains largely consistent across a variety of structures of ETS1 bound in different DNA sequence contexts and in some cases associated with other DNA-binding proteins (e.g., PAX5, RUNX1, etc.). The other major CPD hotspot occurs at position −4/−3 relative to the ETS motif midpoint; however, there are no available ETS1 structures with a dipyrimidine at these positions.

Analysis of a co-crystal structure of GABPA bound to DNA (Fig. 6d) revealed a similar trend, with the GABPA-bound DNA having a favorable distance ($d$) and torsion angle ($\eta$) for CPD formation, particularly at the TC dipyrimidine at position −1/0 (see arrow in Supplementary Fig. 8). To investigate the extent to which these DNA conformations were pre-existing or a direct consequence of GABPA binding, we performed molecular dynamics simulations that modeled the ETS motif-2 in the *RPL13A* promoter fragment (see Fig. 6a) in the presence and absence of bound GABPA protein. The simulations indicated that GABPA binding narrows the ensemble of DNA conformations within the ETS motif around more favorable distances and torsion angles for CPD formation relative to unbound DNA (Fig. 6e). This is particularly apparent for the CPD hotspot at position −1/0 in the ETS motif (i.e., $T_{-1}$-$C_0$ in Fig. 6e). Not only is the distance between the respective C5–C6 double bonds closer in the GABPA-bound simulation, but there is also a significant shift in the distribution of torsion angle toward more favorable values upon GABPA binding. These findings are consistent with our analysis of static structures of ETS1 and GABPA-bound DNA, and show that ETS proteins impose dynamic constraints on the bound DNA that predispose specific dipyrimidine base steps to CPD formation.

## Discussion

Here we have shown that DNA binding by ETS TFs induces a unique UV damage signature, both in cells and in vitro, that is highly correlated with elevated mutation density in melanomas (Fig. 7). In contrast, we find that repair activity is high at ETS binding sites, indicating that increased damage formation, as opposed to repair inhibition, is likely the primary mechanism driving the elevated mutation rates at these sites. Since many ETS family TFs are known oncogenes, and regulate many target genes implicated in cancer development, this mechanism may contribute to the etiology of melanoma and other skin cancers.

Mutation density is highly elevated at two primary locations in ETS binding motifs: (1) the midpoint (position 0) of the ETS motif and (2) position −3 (and −4) relative to the motif midpoint (Fig. 7). The first mutation hotspot has ~15-fold excess of mutations and is associated with ETS binding sites throughout the genome. This mutation hotspot disrupts a highly conserved nucleotide in the core of the ETS motif, which is critical for ETS TFs to bind site-specifically to DNA, based on both in vitro binding experiments and structural studies (e.g.,[24,32]). Moreover, a C-to-T mutation at position 0 in ETS motif has been shown to significantly reduce ETS TFs binding (e.g., *SDHD* promoter[33]), and in some cases alter the expression of neighboring genes[18,27,33]. The second hotspot has >100-fold excess of mutations, but this enrichment is only associated with the subset ETS binding sites that have a lesion-forming dipyrimidine at positions −4/−3 relative to the ETS motif midpoint. Mutations at these positions, which flank the core 5′-TTCC-3′ consensus, are known to reduce ETS binding affinity in vitro[34–36], and have been associated with a decrease in the expression of a reporter gene in

human cells[7]. Some of the most recurrent mutations at these motif positions have been detected in previous studies[7,8,18,26,27]; however, the reason for the extreme mutability of these sites was previously unclear.

A recent study has suggested that lower repair activity at TFBS, including ETS binding sites, is the primary driver of elevated mutation density at these sites[11]. In contrast, we found high repair activity at ETS binding sites, even though we also analyzed CPD XR-seq data at similar time points. This discrepancy can be explained, at least in part, by the inclusion of a large number of discovered motifs in the list of ETS binding sites in the previous analysis[11]. Discovered motifs are novel regulatory sequences significantly associated with ETS ChIP-seq peaks, but which do not match the consensus binding motif[23]. Our analysis indicates that low repair activity is attributable to these discovered motifs, which consist primarily of GC-rich repetitive sequences, while bona fide ETS consensus motifs (known motifs) are associated with high repair activity at the TFBS (Supplementary Fig. 9). There is extensive data indicating that ETS TFs bind to known motifs; however, experimental data indicating that ETS TFs bind to many of these discovered motifs are lacking. Taken together, our results suggest that recurrent mutations at ETS binding sites are primarily a consequence of the unique UV damage signature induced by ETS binding, as opposed to lower repair activity. These findings, consistent with a recent study[6], implicate variable lesion formation as a key contributor to mutation heterogeneity in cancer.

Importantly, we show that ETS binding induces a similar UV damage signature in cells using CPD-seq and with purified ETS1 protein in vitro. While this unique UV damage signature is readily detectable in our CPD-seq data, it has not been previously reported in other UV damage maps. Analysis of available structural data suggests that ETS1 and GABPA binding induce a DNA conformation that is particularly susceptible to undergoing the UV-induced [2+2] cycloaddition reaction to form a CPD lesion. The distance between adjacent C5–C6 double bonds is somewhat reduced in ETS1-bound DNA and the relative torsion angle between these bonds is smaller, particularly at the TC dipyrimidine at position −1/0 relative to the motif midpoint, causing the corresponding bonds to be more favorably aligned for CPD formation to occur. Molecular dynamics simulations indicate that ETS TF binding directly induces these conformation changes in the DNA that predispose it to form CPD lesions. An implication of this model is that in UV-exposed cells, DNA binding by ETS TFs is inherently mutagenic.

In addition to ETS binding sites, our CPD-seq data indicated that CPD formation is elevated at binding sites for subunits of the Nuclear Transcription Factor Y (NFYA and NFYB) (Fig. 2a). This finding is consistent with previous studies that have shown CCAAT boxes bound by NFYA/NFYB have elevated UV damage levels in human cells[17,37]. However, these CPD hotspots are not associated with elevated mutation density in melanoma (Fig. 2c), presumably because elevated CPD levels at NFYA/NFYB binding sites primarily occur at TT dipyrimidine sequences, which are typically not mutagenic. A number of other TFs, particularly those in the c-Fos and c-Jun family, show decreased CPD formation relative to naked DNA controls, indicating that DNA binding by these TFs suppress CPD formation, consistent with our previous study of yeast TFs[14]. Importantly, decreased mCPD formation generally correlates with a reduced mutational burden at TFBS, indicating that UV damage signatures induced by TF binding are an important determinant of mutation frequency in skin cancers.

In summary, we have shown that ETS TFs induce a unique UV damage signature that drives recurrent mutagenesis at ETS binding sites in melanoma. Moreover, we describe a molecular

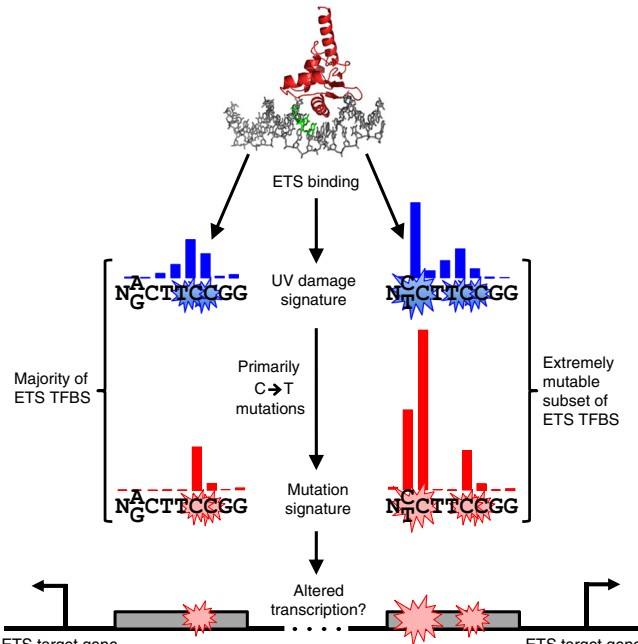

**Fig. 7** ETS TFs induce a unique UV damage signature that drives mutagenesis in melanoma. The extremely mutable subset of ETS TFBS contains a pyrimidine (C or T) at position −4 relative to the ETS motif midpoint, while most ETS TFBS contain a purine (A or G) at this position. Blue bars indicate relative levels of CPD lesions following UV irradiation, while red bars indicate the frequency of mutations in melanoma tumors. 'N' indicates any nucleotide, while stacked nucleotides indicate degeneracy at that position in the ETS binding site motif. ETS binding site mutations may alter transcription of target genes such as *SDHD*. Structure of GABPA bound to DNA (PDB ID: 1AWC) was created using pymol

mechanism responsible for this UV damage signature. ETS TFs are known oncogenes that regulate many genes involved in cell proliferation, differentiation, apoptosis, angiogenesis, and cell migration[38,39]. Mutations at ETS binding sites in the promoters of the *TERT* and *SDHD* genes are known to contribute to the etiology and pathogenesis of melanoma[9,10,27,40]. Thus, it seems likely that the ETS-induced UV damage and mutation signatures promote tumor progression by altering this transcriptional program.

## Methods

**UV irradiation of human cells and genomic DNA isolation**. Telomerase-immortalized normal human fibroblast (NHF1) cells (originally derived by Dr. William Kaufmann, University of North Carolina) were grown to confluence in Dulbecco's modified Eagle's medium (DMEM) containing 10% fetal bovine serum (FBS) at 37 °C and 5% $CO_2$. For UV irradiation, the medium was removed and the residual medium was rinsed off with sterile 1 × Phosphate Buffered Saline (PBS). Cells were then wetted with 2 ml of fresh PBS in a 10 cm petri dish and irradiated with UV-C light (predominantly 254 nm[41]). To map the initial formation of CPDs, NHF1 cells were irradiated with 100 J m$^{-2}$ of UV-C light (or 20 J m$^{-2}$ in Supplementary Fig. 2b) and were harvested immediately after UV treatment.

NHF1 genomic DNA was isolated with Puregene Cell and Tissue kits (158388, Qiagen) or GenElute Mammalian Genomic DNA Miniprep kits (G1N70, Sigma), following the manufacturer's instructions. The isolated genomic DNA was quantified and ~10 μg of purified DNA was used for each sequencing library preparation. The presence of CPDs in DNA isolated from UV-irradiated NHF1 cells was validated by digestion with CPD-specific T4 endonuclease V (Epicentre), followed by alkaline gel electrophoresis. The amount of CPDs initially formed was analyzed using our published protocol[42].

**UV irradiation of naked DNA**. Genomic DNA was first isolated from untreated NHF1 cells, and DNA isolation was conducted as described above. The purified DNA was spotted on a clean microscope cover glass, 10 μl in each drop. The cover glass with DNA spots was irradiated with 80 J m$^{-2}$ UV-C light (mainly 254 nm). UV-irradiated DNA was collected and ~10 μg was sonicated for CPD-seq library preparation.

**CPD-seq library preparation and sequencing**. We adapted the published emRiboSeq method[43] and our yeast CPD-seq protocol[14] to prepare NHF1 and naked DNA CPD-seq libraries. Briefly, the purified NHF1 genomic DNA was sonicated to an average size of ~350 bp using a Bioruptor 300 sonicator (15 cycles, 30″ ON and 30″ OFF), or a Covaris M220 Focused-ultrasonicator. Sonicated DNA fragments were purified, and ~4 μg of DNA was retained. DNA fragments were end-repaired and dA-tailed, and ligated to the double-stranded adaptor DNA trP1[14]. The ligation of trP1 at both 5′ and 3′ ends was validated by PCR with a primer complementary to the trP1 adaptor (primer trP1). Free 3′-OH groups in DNA were blocked with Terminal Transferase (TdT, NEB) and dideoxy ATP (Roche) for 2 h at 37 °C. To validate 3′ end blocking, 10% of DNA was separated, and denatured to ligate any remaining 3′ ends to the A adaptor. The lack of free 3′-OHs was confirmed using PCR with $^{32}$P-labeled primer A and cold primer trP1[14]. The remaining DNA (i.e., 90%) was sequentially incubated with T4 endonuclease V and human apurinic/apyrimidinic (AP) endonuclease (APE1, NEB) to generate new ligatable 3′-OH groups at CPD sites. After removing 5′ phosphates with shrimp alkaline phosphatase (Affymetrix), DNA (~1 μg) was denatured and ligated to A adaptors at 16 °C overnight, using the NEBNext Quick Ligation Module. Distinct barcodes were embedded in the A adaptors to make libraries for different samples and facilitate multiplexed DNA sequencing. One strand in the A adaptors was labeled with biotin, which allows purification of the ligated products with Streptavidin beads (Thermo Fisher Scientific). The DNA strand without the biotin label was released with 0.15 M NaOH. The resulting single stranded DNA was used as the template to synthesize a double-stranded CPD-seq library using primer A as the extension primer. CPD-seq libraries were briefly amplified with PCR (i.e., five cycles), using primers A and trP1. The amplified libraries were size selected with 0.6 volumes of AMpure XP beads, mixed at equal volumes, and sequenced using the Ion Proton platform (Life Technologies). We generated 112 million sequencing reads in total from three independent NHF1 CPD-seq libraries (100 J m$^{-2}$), and 83 million of these CPD-seq reads mapped to a dipyrimidine sequence. Similarly, we sequenced 45 million reads from two independent naked DNA CPD-seq libraries, and 32 million of these CPD-seq reads mapped to a dipyrimidine sequence.

**Oligonucleotides used in CPD-seq**. The following oligonucleotides were used for CPD-seq preparation:

*trP1 adaptor DNA*
trP1-top (5′-CCTCTCTATGGGCAGTCGGTGATphosphorothioate-T-3′)
trP1-bottom (5′-phosphate-ATCACCGACTGCCCATAGAGAGGC-dideoxy-3′)
*Barcoded A adaptor DNA*

A1-top (5′-phosphate-ATCCTCTTCTGAGTCGGAGACACGCAGGGATGAGATGGC-dideoxy-3′),
A1-bottom (5′-biotin-CCATCTCATCCCTGCGTGTCTCCGACTCAGAAGAGGATNNNNNN-C3 phosphoramidite-3′)
A2-top (5′-phosphate-ATCACGAACTGAGTCGGAGACACGCAGGGATGAGATGGC-dideoxy-3′)
A2-bottom(5′-biotin-CCATCTCATCCCTGCGTGTCTCCGACTCAGTTCGTGATNNNNNN-C3 phosphoramidite-3′)
A3-top (5′-phosphate-ATCTCAGGCTGAGTCGGAGACACGCAGGGATGAGATGGC-dideoxy-3′)
A3-bottom (5′-biotinCCATCTCATCCCTGCGTGTCTCCGACTCAGCCTGAGATNNNNNN-C3 phosphoramidite-3′)
A5-top (5′- phosphate-ATCCAGTACTGAGTCGGAGACACGCAGGGATGAGATGGC-dideoxy-3′)
A5-bottom (5′- biotin-CCATCTCATCCCTGCGTGTCTCCGACTCAGTACTGGATNNNNNN-C3 phosphoramidite-3′)
A6-top (5′- phosphate-ATCAGTTCCTGAGTCGGAGACACGCAGGGATGAGATGGC-dideoxy-3′)
A6-bottom (5′- biotin-CCATCTCATCCCTGCGTGTCTCCGACTCAGGAACTGATNNNNNNN-C3 phosphoramidite-3′)
*CPD-seq library confirmation and amplification*
Primer trP1 (5′-CCTCTCTATGGGCAGTCGGTGATT-3′)
Primer A (5′-CCATCTCATCCCTGCGTGTCTCCGAC-3′)

**CPD-seq data analysis**. CPD-seq sequencing reads were trimmed of barcode sequences and the 3′ nucleotide of the sequencing read, and then aligned to the hg19 human genome using the bowtie 2 software[44]. The resulting alignment files were processed with SAMtools[45] and BEDtools[46], and custom Perl scripts were used to identify dinucleotide sequence immediately upstream of the 5′ end of each sequencing read. The dinucleotide sequence on the opposite strand was extracted as a putative CPD lesion. Background reads associated with non-dipyrimidine sequences, which were likely due to incomplete 3′ DNA end blocking or non-specific DNA cleavage by T4 endonuclease V/APE1, were excluded from subsequent analyses. Both positions in the dipyrimidine nucleotide were counted as lesion sites. Three independent CPD-seq experiments mapped CPD lesions in UV-irradiated NHF1 cells (UV cells) and two independent CPD-seq experiments mapped lesions in isolated NHF1 genomic DNA that was UV-irradiated in vitro (UV naked DNA). These biological replicates were combined for most of the analyses. Additionally, in some cases only mutagenic CPD (mCPDs), which are CPD reads associated with TC, CT, or CC dinucleotides, were analyzed.

**Melanoma mutation dataset**. The mutation data from 184 unique tumor samples was acquired from the ICGC data portal DCC data release 20 (https://dcc.icgc.org/releases/release_20/Projects/MELA-AU). Mutations occurring in multiple tumor samples, but from the same patient, were removed to prevent multiple calls of mutations that could have arisen before metastasis and thus inflate our results. Only single base pair somatic mutations were used for subsequent analysis.

**XR-seq data analysis**. The XR-seq data were acquired from ref. [5], and the CPD 1, 4, and 8 h time points (all replicates) were analyzed. We trimmed the sequences by the barcode sequences, and mapped the sequences to the hg19 genome using Bowtie2. To identify the location of the repaired lesion within the XR-seq read, we searched for dipyrimidine sequences starting 6 bases upstream of the 3′ end and ending at 8 bases upstream, based on the findings of a previous study[47]. The first dipyrimidine found within this window was specified as the lesion position. XR-seq reads lacking a dipyrimidine in this sequence window were excluded. The identified XR-seq lesion positions were used for subsequent analyses.

**Analysis of CPD formation and UV mutagenesis in genes**. Gene coordinates were acquired from the CCDS Database (www.ncbi.nlm.nih.gov/projects/CCDS/CcdsBrowse.cgi). RNA-seq data for 470 human melanomas was obtained from (http://gdac.broadinstitute.org/runs/stddata__2016_01_28/data/SKCM/20160128/gdac.broadinstitute.org_SKCM.Merge_rnaseqv2__illuminahiseq_rna-seqv2__unc_edu__Level_3__RSEM_genes_normalized__data.Level_3.2016012800.0.0.tar.gz; project TCGA-SKCM. The average RSEM expression value for each gene was determined among the cohort and used to stratify CCDS genes into quartiles by transcription level. The number of CPD-seq reads and melanoma mutations on the transcribed and non-transcribed strands of in the highest transcribed genes were counted. To account for variations in gene length, each gene was broken into six bins of equal fractional length and counts were made within those bins. In addition to the genes themselves, two 5000 bp bins were analyzed upstream and downstream of the genes. The CCDS sites were also analyzed for their dipyrimidine sequences and all datasets were normalized to them to account for sequence context.

**TFBS coordinates**. TFBS coordinates identified by ENCODE[23], representing binding site motif matches associated with ChIP-seq peaks for the corresponding TF, were derived from ref. [48] and downloaded from http://funseq.gersteinlab.org/

data/ (file: ENCODE.tf.bound.union.bed). Only TFBS associated with a known motif were retained; all TFBS associated with discovered motifs, which often were GC-rich DNA sequences unrelated to the known TF consensus binding sequence, were excluded. Next, overlapping TFBS associated with the same TF were merged into a single binding site, and then intersected with a set of defined upstream promoter regions (up to 2500 bp upstream of the TSS[11]), to obtain a set of merged promoter-proximal TFBS. Promoter-proximal TFBS that overlapped with melanocyte DNase I hypersensitivity sites (DHS) were considered active TFBS[11]. DHS data were from the Epigenome Roadmap Project[49] and downloaded from http:// egg2.wustl.edu/roadmap/data/byFileType/peaks/consolidated/narrowPeak/ (file: E059-DNase.hotspot.fdr0.01.peaks.bed.gz). TFBS associated with blacklisted regions (Duke and DAC) were excluded from this analysis[11]. In total, there were 26,213 active promoter-proximal TFBS representing 82 TFs and 60,596 inactive promoter-proximal TFBS representing 82 TFs.

**Analysis of CPD-seq reads and mutation density at TFBS.** We analyzed CPD-seq lesions located within 100 bp of the midpoint of a TFBS (either active or inactive promoter-proximal TFBS). For the analysis described with Figs. 1–3 (and related figures), the midpoint of each TFBS was determined by averaging the start and end positions of the binding site and, if necessary, rounding up to the nearest integer. In cases where multiple TFBS had the same midpoint, only the first instance was included in the analysis. The average number of CPD lesions at each position relative to the TFBS midpoint was calculated by counting the number of CPD-seq reads associated with each position (either for all dipyrimidine sequences or just mCPDs) and dividing by the total number of unique TFBS included in the analysis. This analysis was performed both for the UV-irradiated NHF1 cell datasets and UV-irradiated naked DNA. Since the pyrimidine frequency in the two DNA strands is negatively correlated, CPD lesions on both DNA strands were combined for all of our analysis, so as to mitigate the potential effects of DNA strand bias on CPD formation at TFBS. A similar analysis was used to analyze the pattern of mutation density, derived from 184 melanoma tumors[18], that was adjacent to TFBS. CPD enrichment at core TFBS (−4 to +4 bp relative to TFBS midpoint) or flanking DNA was calculated by taking the scaled ratio of CPD lesions in cells relative to naked DNA. The scaling factor was calculated based on the total number of CPD reads in promoter-proximal regions in UV-irradiated cells relative to UV-irradiated naked DNA. The CPD enrichment values were scaled so that overall cells/naked DNA ratio in promoter-proximal regions was set to 1. TFs whose set of TFBS contained fewer than five CPD lesions in either the UV-irradiated cells or naked DNA datasets were excluded from the analysis shown in Fig. 2. CPD lesions associated with dipyrimidines on the boundary of the TFBS core (i.e., −5/−4 bp or +4/+5 bp relative to the TFBS midpoint) were counted as 0.5 lesions for the total lesion count (x-axis of Fig. 2).

The expected mutation density adjacent to TFBS was determined by first calculating the mutation frequency for each trinucleotide context (e.g., TCA, where the C is mutated), among the 184 melanoma tumor samples. For the analysis of promoter-proximal TFBS (active and inactive), the trinucleotide mutation frequencies were calculated from the same set of promoter-proximal regions (e.g., 2500 bp upstream of the TSS). The expected mutation density at each position adjacent to a TFBS (e.g., from −100 to +100 bp) was given by the trinucleotide context of the DNA sequence at that position. These expected mutation densities were then summed at each position for all unique TFBS, in a similar manner as for the observed mutation density, and then divided by the total number of unique TFBS. Mutation enrichment at core TFBS (−4 to +4 bp relative to TFBS midpoint) was calculated by taking the ratio of observed mutations in the 184 melanoma tumor samples relative to expected mutations (see above) for the core TFBS. Only TFs with at least 1 mutation (cumulative) in a core TFBS and at least 50 active promoter-proximal TFBS were included in the analysis shown in Fig. 2.

For high-resolution CPD-seq and mutation density analysis of ETS family TFBS (i.e., Fig. 4 and S5), the binding site midpoint was determined in a similar manner, although in this case taking into account the DNA strand information of the binding site (i.e., midpoint was rounded up for TFBS on the plus strand, and rounded down for TFBS on the minus strand). The midpoint of each ETS family TFBS was shifted by a predetermined offset, depending on the particular ETS motif, so that all binding sites were aligned at the central TTCC core consensus, with the midpoint corresponding to the underlined C nucleotide. ETS motifs were oriented so that the strand containing the TTCC consensus sequence was in the 5′ to 3′ orientation, and CPD lesions on both DNA strands were combined in the analysis. Mutation density was analyzed relative to unique ETS family TFBS midpoints, as described above. For high-resolution analysis of the CPD-seq reads (i.e., Fig. 4 and Supplementary Fig. 5), the location of the lesion was given as the fractional midpoint of the two nucleotides comprising the lesion (e.g., a CPD lesion comprised of nucleotides 50 and 51 would be assigned a position of 50.5), and analyzed relative to unique ETS family TFBS midpoints, as described above.

**Analysis of CPD formation at ETS binding sites in vitro.** Recombinant TF ETS1 (residues 280–440, ETS1ΔN280) was expressed in *Escherichia coli* and purified by immobilized metal affinity chromatography and cation exchange chromatography[50]. Oligonucleotides containing ETS motifs derived from *RPL13A* and *SDHD* promoters were synthesized and PAGE-purified by IDT (Integrated DNA Technologies). Oligonucleotides sequences were RPL13A-FWD (5′-Biotin-

GGTCCAACCGGAAGAATGTCCGGATTGGAC-3′); RPL13A-RVS (5′-GTCCAATCCGGACATTCTTCCGGTTGGACC-3′); SDHD-FWD (5′-CTCGACTTCCGGTTCACCCAGCATTTCCTCTTCCCTGTT-3′); SDHD-RVS (5′- Biotin-AACAGGGAAGAGGAAATGCTGGGTGAACCGGAAGTCGAG-3′). Additionally, mutant SDHD oligonucleotides with point mutations at the consensus ETS motifs were also synthesized. The mutant DNA sequences were SDHD-mt1-FWD (5′-CTCGACTTGAGGTTCACCCAGCATTTCCTCTTCCCTGTT-3′); SDHD-mt1-RVS (5′-Biotin-AACAGGGAAGAGGAAATGCTGGGTGAACC TCAAGTCGAG-3′); SDHD-mt2-FWD (5′-CTCGACTTCCGGTTCACCCAG-CATTTGATCTTCCCTGTT-3′); SDHD-mt2-RVS (5′-Biotin-AACAGGGAA-GATCAAATGCTGGGTGAACCGGAAGTC GAG-3′). (Note: mutations are underlined). After annealing of the paired oligonucleotides, the DNA strand with the consensus ETS motif sequence (TTCC) was labeled with [γ32P]-ATP (Perkin Elmer) and T4 polynucleotide kinase (PNK, New England Biolabs) at the 5′ terminus. The complementary DNA strand was prelabeled with biotin during oligonucleotide synthesis and thus was not radiolabeled by PNK.

Binding between ETS1 protein and the labeled DNA fragments was determined by electrophoretic mobility shift assays[51]. Briefly, the reaction mixtures (40 μl) were in the binding buffer (25 mM Tris-Cl (pH 7.9), 10% glycerol, 6 mM MgCl2, 0.5 mM EDTA, 60 mM KCl, 0.5 mM DTT and 200 μg/ml BSA) with radiolabeled DNA (4 pm) and ETS1 protein (ranging from 0 to 32 pm). Following a 40-min incubation on ice, 4 μl of binding products were loaded to a native polyacrylamide gel (8%) to confirm binding, and the remainder of the binding products was irradiated with 254-nm UV light.

For UV irradiation, the binding products were spotted on a clean microscope cover glass, 9 μl for each spot. The cover glass was placed on ice and irradiated with a handheld UV lamp (Spectroline, Model ENF-240C). *SDHD* promoter DNA fragment was irradiated with 750 J m$^{-2}$ of UV while the relatively short *RPL13A* promoter DNA fragment was irradiated with 1000 J m$^{-2}$ of UV light. The UV-treated binding products were collected and protein was removed by phenol-chloroform-isoamyl alcohol (25:24:1) extraction. DNA was precipitated with ethanol, and the purified DNA was subsequently incubated with ~2 U of purified T4 endonuclease V at 37 °C for 1 h to generate single strand breaks at CPD sites. As a control, DNA fragments without UV treatment were also incubated with the same amount of T4 endo V in parallel.

The digestion products (10 μl) were mixed with the same volume of formamide (Sigma-Aldrich) and heated at 75 °C for 5 min, and loaded to a denaturing urea polyacrylamide sequencing gel. Sequencing gel electrophoresis was conducted as described previously[52]. The gel was exposed to a PhosphorImager screen (Molecular Dynamics) and radioactive signal was detected by a Typhoon FLA 7000 laser scanner (GE Healthcare Life Sciences). Gel quantification was conducted with the ImageQuant TL software (GE Healthcare Life Sciences).

To determine the sizes of T4 endo V digestion products, multiple truncations at the 3′ termini of the *RPL13A* and *SDHD* promoter fragments were designed to show the precise sizes of CPD bands on the gel. The truncations were only designed for the DNA strand containing the ETS consensus sequence (TTCC) and the resulting single-stranded oligonucleotides were 5′ end labeled and loaded to the sequencing gels side-by-side with T4 endo V-treated samples. Supplementary Fig. 10 shows an uncropped gel image containing the molecular weight/size markers for the gel shown in right panel of Supplementary Fig. 6c.

**Structural analysis of ETS1- and GABPA-bound DNA.** Structural analysis was performed using structures of ETS1-bound to a canonical ETS motif (i.e., [A/T] TCC). The PDB IDs for these structures are: 1K79, 1K7A, 2NNY, 2STW, 3MFK, 3RI4, 3WTS, 3WTT, 3WU1, 4L0Y, 4L0Z, 4L18, and 4LG0. These structures were used to calculate the distance (d) between the midpoints of C5-C6 bonds of adjacent pyrimidines, as well as the torsion angle (η) between the adjacent C5-C6 bonds[29]. For the analysis of the structure of GABPA-bound to DNA, PDB ID 1AWC was analyzed in a similar manner.

**Molecular dynamics simulations.** All-atom molecular dynamics simulations were performed in the GROMACS 2016.4 environment. An initial B-form duplex encoding d-(CGGACATTCTTCCGGTTGGACC) was constructed with 3dna[53]. A second model was generated by docking the DNA to the ETS domain of GABPA (murine residues 320 to 429) using the co-crystallographic structure 1AWC[32] as template. The system was solvated in explicit TIP3P water and 0.15 M NaCl in a dodecahedral box sized 10 Å larger than the longest (axial) dimension of the DNA. All simulations were carried out at an in silico temperature and pressure of 298 K (modified Berendsen thermostat) and 1 bar (Parrinello-Rahman ensemble)[54] using the amber99/parmbsc1 force field[55,56]. All bonds were constrained using LINCS. After the structures were energy-minimized by steepest descent, the NVT ensemble was equilibrated at 298 K for 100 ps to thermalize the system, followed by another 100 ps of equilibration of the NPT ensemble at 1 bar and 298 K. The NPT ensemble at 298 K was simulated for 200 ns; trajectories were recorded every 10 ps. Convergence of DNA dynamics was confirmed from RMSD of all DNA atoms. For unbound and GABPA-bound DNA, the distance between C5 and C6 and torsion angle η between pyrimidine/pyrimidine dinucleotide steps between positions −5 to +5 were computed from the final 100 ns (10,000 frames) and binned (at increments of 0.05 Å for distances; 1° for dihedrals) for frequency analysis.

**Data availability**. The CPD-seq data have been deposited in the Gene Expression Omnibus (GEO) database, www.ncbi.nlm.nih.gov/geo (accession number GSE103487). The data are available upon request.Computer code used to analyze the CPD-seq and mutation datasets as well as molecular dynamics trajectories are also available upon request.

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

## Acknowledgements

The authors are grateful to Wei Wei Du and Mark Wildung for technical assistance with Ion Proton sequencing and Drs. Dmitry Gordenin and Amelia Hodges for helpful comments and suggestions. The authors thank Dr. Barbara Graves for providing purified ETS1 protein. The authors thank Tristan Watts, Reilly Bucy, and Laina Wyrick for assistance with bioinformatics analysis. The authors are also grateful to the International Cancer Genome Consortium and in particular the Skin Cancer—AU project for the sequencing of the human melanoma genomes and making the somatic mutation calls publicly available. Some results within this manuscript are based upon RNA-seq analysis conducted by the TCGA Research Network cutaneous melanoma working group. Funding for this research was provided by grants from NIEHS (R01ES002614 to J.J.W. and M.J.S., R21ES027937 to J.J.W. and S.A.R., and R00ES022633 to S.A.R), NCI (R01CA218112 to S.A.R.), NIAID (T32AI07025 to S.L.), NHLBI (R21HL129063 to G.M. K.P), NSF (MCB 15451600 to G.M.K.P.), and an internal grant from the Washington State University College of Veterinary Medicine (to J.J.W.). S.E. is a Molecular Basis of Disease Fellow at Georgia State University.

## Author contributions

P.M., A.J.B., S.L., G.M.K.P., M.J.S., S.A.R, and J.J.W. designed the research and interpreted results; P.M. performed CPD-seq and ETS1 in vitro experiments; P.M., A.J.B. and J.J.W. analyzed the CPD-seq data; A.J.B., J.J.W, and S.A.R. analyzed melanoma mutations and XR-seq data; S.E. purified the ETS1 protein and G.M.K.P. performed molecular dynamics simulations; S.L. and J.J.W. analyzed the ETS1-DNA structures; P.M., A.J.B., G. M.K.P., M.J.S., S.A.R, and J.J.W. wrote the paper.

## Additional information

**Competing interests:** The authors declare no competing interests.

