## [Peer Review File · Nature Communications]

Reviewers' Comments:

Reviewer #1:

Remarks to the Author:

The manuscript by Mao et al. describes the use of an improved high-throughput sequence method that creates single-nucleotide resolution maps of CPD lesions, bypassing the immunoprecipitation of thymidine dimer-specific dipyrimidines, to demonstrate that UV-induced noncoding cyclobutane pyrimidine dimer (CPD) mutations found in melanomas are enhanced in active transcription factor binding sites, principally associated with E26 transformation-specific transcription factor (ETS) binding. They go on to show that bound ETS factors enhance mutagenesis at specific CPD hotspots, at least in part through binding-induced local perturbation in the DNA structure, despite relatively high nucleotide excision repair activity at these sites. Although enhanced mutation frequencies at transcription factor binding sites has been reported previously, this work provides new mechanistic insight into how specific transcription factors can drive UV-mediated mutagenesis at specific binding sites, and drive the development of skin cancer. A few points that if addressed would strengthen this paper.

According to this study, mutagenesis is enhanced at sites where ETS1 and related family members bind. Perhaps I missed this, but can ETS factors still bind to these UV-mutated binding sites at the same affinity? It is important to be clear about this point.

Several experimental studies were performed in fibroblasts. It is not clear that these same mechanisms apply to melanocyte or keratinocyte cell types, which would be the cells most vulnerable to UV mutation-associated cancer. Some validation should accompany the fibroblast studies.

Reviewer #2:

Remarks to the Author:

This exciting and well written study by an outstanding team of investigators tests the hypothesis that transcription factor "binding may alter the rate at which UV lesions form in DNA". While this team has begun to investigate this question in yeast cells (their recent paper in Proc Natl Acad Sci U S A 113,9057-62, 2016.) this problem has not been addressed in any significant way in the context of the environment of mammalian cell nucleus. Thus, this study will have an important lasting impact on the field. In order to fully address this question, and what sets it apart from previous work by the Sancar and Morrison laboratories is the development and validation of a new method, CPD-seq for analyzing cyclobutane pyrimidine dimer- induced damage at base pair resolution across the mammalian cell genome. The strength of this assay is that it does not require immunoprecipitation of CPD, which in the past probably underestimated the frequency of CPD at sites other than TTs. Using this new tool, the authors nicely demonstrate that "ETS TFs induce a unique signature of UV damage hotspots at specific locations in their binding motif, both in cells and in vitro". The experiments for the most part are well described and carefully repeated to gain strong statistical significance. Moving from an initial observation in mammalian cells to validation with a purified in vitro system is a strength of this study. 7. To this end, Figures 5 & 6 add a significant amount to this study and provide a strong validation of their major hypothesis. Also comparing signature mutations in melanomas to the frequency of CPD at TF is another important strength of this work. Finally, the addition of extensive supplementary data adds significantly to this work. Overall this is an outstanding study that will have lasting impact on the field once published. The following suggestions are to improve the overall impact of the study.

1. The authors down play the significance of the frequency of 6-4 photoproducts in the introduction (top of page 3). The authors indicate, "to a much lesser extent, pyrimidine (6-4) pyrimidone photoproducts (6-4PPs) at dipyrimidine sequences in DNA." The authors should avoid this vague term and indicate that depending on the sequence the frequency of 6-4 photoproduct can range anywhere from 10% to equal numbers. This also raises the question as what happens

when there is a 6-4 photoproduct within the same sequence as a CPD – while rare this could happen.

2. Again top of page 3, Citations of previous work. Overall the authors do a OK job, but miss the opportunity on page 3 to include references to recent Sancar papers and a recent paper from the Morrison lab, EMBO J. 2017 Oct 2; 36(19):2829-2843. They do cite these papers later, but not in this early context.

3. The authors do a fair job of detailing the various doses of UV-C used in their experiments in the methods, but to make it easier for the reader, it would be helpful to indicate the doses used in the results sections and also the figure legends whenever possible. In the methods the authors state: "NHF1 cells were irradiated with 20 J/m² or 100 J/m² UV and were harvested immediately after UV treatment." However which dose goes with which figure is not always clear. This is especially important in Figure legend 1 that sets the bar for the rest of the study. And later in Figure 3 where they are making direct comparisons to mutation frequencies – can one dose be used to extrapolate to mutation frequencies?

4. What is the relative ability to detect a lesion with this sequencing method, i.e. what is the signal to noise and the depth of the sequencing – how high is the resolution? This needs to be clearly stated early in this present paper. Perhaps they did a dose response in their previous work-up of yeast cells (PNAS 2016). Understanding these questions is essential for drawing all the following conclusions.

5. On page 9 the authors make the strong statement, "Elevated UV damage, not lower repair activity, promotes mutagenesis at ETS binding sites." However, they used previous data from one study, Nature 532, 264-7 (2016) to support this claim. Why did they not include recent data from the Sancar laboratory as well? This again occurs in the discussion on page 18. The authors should also cite the Morrison EMBO J as they make a similar claim in their study. Overall this is sure to be controversial so the stronger the argument that can be made in this study the better.

6. The gray and black lines in Figure 4 are difficult to see clearly – a better color scheme is advised.

Reviewer #3:

Remarks to the Author:

Summary:

A long-standing question is why tumor mutations occur where they do. The answer was long assumed to be selection for altered protein function or regulation, but in the past few years factors such as chromatin and replication timing have been shown to also influence the likelihood of causing a driver mutation. Recently, several papers bioinformatically identified transcription factor binding sites (TFBS) as determinants of mutation hotspots in melanoma (notably refs 6 and 9), and at least one prominent paper (ref 9) ascribed this effect to reduced repair at TFBS. The latter paper's observation was a little puzzling because the repair defect was only a few-fold, resulting in reducing an elevated repair in the regions flanking the TFBS back to a repair level typical of the genome as a whole; why would this give that site more than the genome-wide density of mutations?

The current paper is a pivotal contribution to our understanding. It proceeds on two fronts: a closer look at the TFBS sequences involved, and direct measurement of the frequency of UV-induced cyclobutane dimers (CPD) at single bases. Given the prior literature, the CPD measurements are the most novel contribution. The authors find that if they focus on canonical TFBS, rather than all sequences immunoprecipitated by TFBS as prior papers did (a point that is buried in the Discussion, perhaps out of diplomacy): A subclass of TFBS -- ETS1 -- are associated with melanoma mutation hotspots; these sites are hotspots for CPD formation; a 16-fold elevation in CPDs accompanies a 125-fold increase in mutations; repair is elevated at these sites, not suppressed, so the mutation site is chosen by having more initial damage not less repair; melanoma mutations are correlated only with ETS1 sites having a TC or CC sequence, allowing the UV signature C->T mutation; this correlation is mechanistically confirmed using DNA bound to ETS1 and irradiated in vitro, but is prevented by mutant ETS1 that cannot bind to one of the two

hotspot sites; these results are nicely explained by the biophysics of CPD formation, which requires certain geometries (bond distances and angles) that are in fact created in DNA-ETF1 crystal structures; and molecular modeling shows that ETS1 shifts the thermally induced range of DNA conformations into the required range of geometries. QED.

Critique:

This is a very careful and thoughtful study, with nice controls such as in vitro validation and, within that validation, mutants that abolish the behavior being tested. The writing is clear.

Some minor points need to be addressed:

Scientific –

1. The two DNA strands are combined for the analysis. The authors' previous paper explains why they do this, but it would help the reader to explain it in the Supplemental here, too.
2. By eyeball, the CPD motif seems to usually or always be on the template strand. Are there examples on the non-transcribed strand that would serve to indicate the role of transcription-coupled repair in converting the CPD to a mutation?
3. The read depth of the experiments needs to be stated. This also bears on the "sample size" reporting requirement.
4. The Introduction states that prior studies using CPD immunoprecipitation suffered from the limitation that they would only precipitate T-containing CPDs. But the original Mori '91 paper states that the TDM-2 antibody binds TT and CT dimers. If there is a later reference showing this not to be the case, it should be cited.
5. A hypothetical point, for the authors to think about. They are careful to say that the CPD is the cause of these melanoma mutation hotspots rather than "reduced repair". Intentionally or not, they don't say repair is not important. This leaves open the possibility that the mutation hotspot is related to repair in an additional way – arising from error-prone repair of the CPD at this site of ETF1-altered DNA conformation. Error-prone repair (as opposed to replicational bypass) is an old idea that I haven't heard discussed lately.

Writing –

1. The authors would be better served in the Abstract and Introduction by talking about "mutations in regulatory regions" rather than "non-coding mutations". To readers not in the field, and maybe in it, "non-coding mutations" sounds like "silent mutations" that don't change the amino acid.
2. The logical flow would be clearer if Fig 1 were only about narrowing down the specificity of the TFBS and then Fig 2 showed CPDs. This is because in the current Fig 1, there is a broad mutation peak yet a narrow CPD peak. It's not until Fig 3F that this all becomes clear to the reader, and then only if s/he heeds the warning that the axis scale is different on the inset. I sympathize with the desire to get to the CPD data as soon as possible, so perhaps another approach would be keep the current figures but to call attention to the apparent discrepancy in peak width and say "and therefore we investigated the TFBS specificity in greater detail".
3. p9 line 11. "mCPD formation at the center of the active TFBS"
4. p12 line 13. Table S1 seems to now be S3.

5. p14 line 5 up. "... panel, site 1)."
6. p15 line 4. "... panel, sites 2, 3)"
7. p29 line 11. "complementary"
8. Fig 1 B-D What is the unit on the y axis? Fraction, I assume, but what exactly?
9. p41 Fig 4 legend. The "lower panel" seems to now be Supplementary Tables S1 and S2.

Response to Reviewers:

Referee #1:

1. "According to this study, mutagenesis is enhanced at sites where ETS1 and related family members bind. Perhaps I missed this, but can ETS factors still bind to these UV-mutated binding sites at the same affinity? It is important to be clear about this point."

RESPONSE: The core ETS binding motif, consisting of TCC (GGA on the opposite strand), is invariant across all ETS binding sites, and is critical for ETS proteins to bind DNA, based on both *in vitro* binding experiments and structural studies^{1,2}. A C-to-T mutation at position 0 relative to the ETS midpoint (i.e., TCC), which is enriched in melanoma, is expected to disrupt ETS transcription factor binding affinity to nonspecific levels. Indeed, a recent study shows that a C-to-T mutation at this location, which mimics the UV-induced mutation in melanomas, significantly reduces binding of GABPA (an ETS family member) to the ETS motif in the SDHD promoter *in vitro*³. We have included a detailed discussion of this important point on page 18 of the revised manuscript.

2. "Several experimental studies were performed in fibroblasts. It is not clear that these same mechanisms apply to melanocyte or keratinocyte cell types, which would be the cells most vulnerable to UV mutation-associated cancer. Some validation should accompany the fibroblast studies."

RESPONSE: We show using our CPD-seq method that UV-induced CPD lesions are elevated at ETS binding sites in UV-irradiated fibroblasts. We further show using an *in vitro* model system that purified ETS1 protein stimulates UV-induced CPD lesions at ETS binding sites *in vitro*, and describe the molecular mechanism likely responsible for elevated CPD formation at ETS binding sites. These *in vitro* studies validate our findings in fibroblasts, and further demonstrate that ETS binding alone, in the absence of any additional, cell-type specific co-factors, induces a unique UV damage signature. Given the conserved DNA binding mechanism by ETS transcription factors across different cell types (and in different species), we expect that the same UV damage signature will occur in melanocyte or keratinocyte cells. We plan to map CPD lesions in these cell types in our future studies, but we believe these experiments are beyond the scope of the current manuscript.

Referee #2:

1. "The authors down play the significance of the frequency of 6-4 photoproducts in the introduction (top of page 3). The authors indicate, 'to a much lesser extent, pyrimidine (6-4) pyrimidone photoproducts (6-4PPs) at dipyrimidine sequences in DNA.' The authors should avoid this vague term and indicate that depending on the sequence the frequency of 6-4 photoproduct can range anywhere from 10% to equal numbers. This also raises the question as what happens when there is a 6-4 photoproduct within the same sequence as a CPD – while rare this could happen."

RESPONSE: We have modified the sentence (see page 3) to read: “UV light induces the formation of cyclobutane pyrimidine dimers (CPDs) and, to a lesser extent, pyrimidine (6-4) pyrimidone photoproducts (6-4PPs) at dipyrimidine sequences in DNA.” We did not feel it was necessary to indicate the exact fold-difference in lesion formation between CPDs and 6-4PPs, since this is described in much more detail in the cited reference, and is not the focus of our study. Since CPD lesions are overall more abundant (~3-4-fold higher than 6-4PPs following UV-C irradiation⁴) and are repaired more slowly than 6-4PPs, and because it has been long recognized that CPDs are the major mutagenic lesion in skin cancers (e.g., ⁵), we mapped just the CPD lesions in our study.

It is possible (albeit very unlikely) that a nearby 6-4PP could impact our ability to map a CPD lesion using CPD-seq. If the 6-4PP were within ~150 bp to 5' side of the CPD lesion on the same DNA strand, we would not be able to PCR amplify and sequence the DNA fragment associated with that particular CPD lesion. However, this is unlikely to have a significant effect on our CPD-seq data, since the UV doses used in our study yielded a relatively low density of UV lesions (i.e., for 100 J/m², there are roughly 0.8 CPD lesions per kb, based on T4 endo V digestion and alkaline gel electrophoresis of the irradiated DNA sample; 6-4PPs would be even lower than this).

2. "Again top of page 3, Citations of previous work. Overall the authors do a OK job, but miss the opportunity on page 3 to include references to recent Sancar papers and a recent paper from the Morrison lab, EMBO J. 2017 Oct 2;36(19):2829-2843. They do cite these papers later, but not in this early context."

RESPONSE: We have now included the references mentioned by the reviewer near the top of page 3 of the revised manuscript.

3. "The authors do a fair job of detailing the various doses of UV-C used in their experiments in the methods, but to make it easier for the reader, it would be helpful to indicate the doses used in the results sections and also the figure legends whenever possible. In the methods the authors state: “NHF1 cells were irradiated with 20 J/m² or 100 J/m² UV and were harvested immediately after UV treatment.” However which dose goes with which figure is not always clear. This is especially important in Figure legend 1 that sets the bar for the rest of the study. And later in Figure 3 where they are making direct comparisons to mutation frequencies – can one dose be used to extrapolate to mutation frequencies?"

RESPONSE: To clarify, most CPD-seq experiments were conducted with the dose of 100 J/m² UV-C. The only experiment that used 20 J/m² was shown in Supplemental Fig. S2B. We have modified the sentence mentioned in the Methods section (page 21) to make this more explicit. We've also clarified the UV doses used in the legend to Figure 1 and in the results text.

Data in Figure 3 are used to highlight the similar trends for CPDs and mutations, as both peak near the midpoint of ETS binding sites. The CPD yields generated at 100 J/m² are not meant to be directly extrapolated to mutation frequency in sequenced melanoma tumors. We chose 100J/m² as a representative UV dose for CPD-seq

experiments, as it is difficult to estimate the average UV dose for the 184 sequenced melanoma tumors.

4. "What is the relative ability to detect a lesion with this sequencing method, i.e. what is the signal to noise and the depth of the sequencing – how high is the resolution? This needs to be clearly stated early in this present paper. Perhaps they did a dose response in their previous work-up of yeast cells (PNAS 2016). Understanding these questions is essential for drawing all the following conclusions."

RESPONSE: In the revised manuscript, we have expanded the bottom paragraph of page 6 and the top of paragraph of page 7 in the results section to more clearly describe the human CPD-seq data, as requested by the reviewer. The CPD-seq method maps CPD lesions at single nucleotide resolution, as discussed in more detail in Mao et al., *PNAS* 2016. Analysis of the CPD-seq data shown in Supplemental Fig. S1B indicates that there is ~10-fold enrichment of sequencing reads associated with dipyrimidine sequences in the UV-irradiated sample (100 J/m²) relative to a matched unirradiated control, providing an estimate of the signal to noise ratio. In our experience, this enrichment ranges from 3-fold to 11-fold, depending upon the dipyrimidine sequence and the individual experiment. In total, 74% of the total sequencing CPD-seq reads for the UV-irradiated NHF1 cells are associated with lesions at the 4 dipyrimidine sequences (i.e., TT, TC, CT, and CC), and the remaining 26% are associated with the 12 non-dipyrimidine dinucleotide sequences. We used CPD-seq to map CPD lesions in UV-irradiated human cells in three independent experiments, which in total comprise 83 million CPD-seq reads that mapped to a putative lesion-containing dipyrimidine sequence (out of 112 million total sequencing reads). The sequencing depth is mentioned on page 7 and page 23 of the revised manuscript.

5. On page 9 the authors make the strong statement, "Elevated UV damage, not lower repair activity, promotes mutagenesis at ETS binding sites." However, they used previous data from one study, *Nature* 532, 264-7 (2016) to support this claim. Why did they not include recent data from the Sancar laboratory as well? This again occurs in the discussion on page 18. The authors should also cite the Morrison *EMBO J* as they make a similar claim in their study. Overall this is sure to be controversial so the stronger the argument that can be made in this study the better.

RESPONSE: The *Nature* 532, 264-7 (2016) from the Lopez-Bigas laboratory (Sabarinathan et al.) was the first paper to show generally elevated mutation rates at TFBS in melanoma tumors. Their bioinformatics analysis of the XR-seq data published by the Sancar laboratory suggested that elevated mutation rates at TFBS were due to inhibition of repair activity. In the revised manuscript, we have included a reference to a review from the Sancar group that also mentions this claim (see page 10), as suggested by the reviewer. Our own re-analysis of XR-seq data from the Sancar laboratory (reference 5 in the manuscript) indicates that there is elevated NER activity at the bona fide ETS binding sites at multiple repair time points.

We now mention the Morrison *EMBO J* study on page 19 of the revised manuscript, as suggested by the reviewer. The recent paper from the Sancar group mapping CPD

lesion formation in human cells⁶ did not analyze CPD formation at ETS1, ELK4, or GABPA binding sites, nor did they correlate variations in CPD formation with mutation rates, so we did not cite this study to support our findings.

6. The gray and black lines in Figure 4 are difficult to see clearly – a better color scheme is advised.

RESPONSE: We have modified symbols in figure 4 so that the naked DNA line is given as rectangles and the line is dashed, so that it can be more clearly distinguished from black line. We experimented with other color schemes, but we found them more distracting. Moreover, the black and gray lines are used to represent CPD lesion in cells and naked DNA, respectively, in other figures, so we have kept this color scheme for consistency.

Referee #3:

1. The two DNA strands are combined for the analysis. The authors' previous paper explains why they do this, but it would help the reader to explain it in the Supplemental here, too.

RESPONSE: We added an explanation in the revised Methods section (see pages 28 and 29) explaining why the two DNA strands are combined, as suggested by the reviewer.

2. By eyeball, the CPD motif seems to usually or always be on the template strand. Are there examples on the non-transcribed strand that would serve to indicate the role of transcription-coupled repair in converting the CPD to a mutation?

RESPONSE: Our data indicate that the increased damage formation on the strand that contains the consensus TTCC sequence is likely the major mechanism that drives mutation elevation at ETS binding sites. Since our current CPD-seq data does not include repair time points, we are unable to examine the kinetics of CPD removal between the two DNA strands at ETS motifs, but this is an interesting suggestion and is worthy of further investigation in future studies.

3. The read depth of the experiments needs to be stated. This also bears on the "sample size" reporting requirement.

RESPONSE: Our three independent experiments mapping CPD lesions in UV-irradiated NHF1 cells generated 112 million CPD-seq reads. The sequencing depth is now mentioned on page 7 and page 23 of the revised manuscript.

4. The Introduction states that prior studies using CPD immunoprecipitation suffered from the limitation that they would only precipitate T-containing CPDs. But the original Mori '91 paper states that the TDM-2 antibody binds TT and CT dimers. If there is a later reference showing this not to be the case, it should be cited.

RESPONSE: We have revised this statement in the Introduction section (see page 4 of the revised manuscript) to clarify that the TDM-2 antibody may specifically precipitate (and thus enrich for) TT and CT dimers. Neither of these dipyrimidines are prime sites of UV-induced mutation in melanoma.

5. A hypothetical point, for the authors to think about. They are careful to say that the CPD is the cause of these melanoma mutation hotspots rather than "reduced repair". Intentionally or not, they don't say repair is not important. This leaves open the possibility that the mutation hotspot is related to repair in an additional way – arising from error-prone repair of the CPD at this site of ETF1-altered DNA conformation. Error-prone repair (as opposed to replicational bypass) is an old idea that I haven't heard discussed lately.

RESPONSE: The idea that error-prone repair may occur during repair synthesis and thus contribute to elevated mutation rates is indeed interesting. However, it is unclear whether the error rate for repair synthesis would be particularly high at ETS binding sites. We hope to address this possibility in future studies by analyzing the frequency of UV-induced mutations at ETS binding sites in XPC patients, who are deficient for the GG-NER pathway.

Writing –

1. The authors would be better served in the Abstract and Introduction by talking about "mutations in regulatory regions" rather than "non-coding mutations". To readers not in the field, and maybe in it, "non-coding mutations" sounds like "silent mutations" that don't change the amino acid.

RESPONSE: To avoid confusion, we have either altered the text to clarify that we are referring to mutations at TFBS, or specified that the mutations are occurring at regulatory elements.

2. The logical flow would be clearer if Fig 1 were only about narrowing down the specificity of the TFBS and then Fig 2 showed CPDs. This is because in the current Fig 1, there is a broad mutation peak yet a narrow CPD peak. It's not until Fig 3F that this all becomes clear to the reader, and then only if s/he heeds the warning that the axis scale is different on the inset.

I sympathize with the desire to get to the CPD data as soon as possible, so perhaps another approach would be keep the current figures but to call attention to the apparent discrepancy in peak width and say "and therefore we investigated the TFBS specificity in greater detail".

RESPONSE: To clarify the discrepancy in peak width between CPD and mutation data, we have now added the text suggested by the reviewer at the top of page 8: "However, the mutation peak at active TFBS is broader than the CPD peak; therefore, we

investigated the impact of individual TFs on CPD formation and mutagenesis in more detail."

3. p9 line 11. "mCPD formation at the center of the active TFBS"

RESPONSE: This has been corrected.

4. p12 line 13. Table S1 seems to now be S3.

RESPONSE: This has been corrected to Table S3.

5. p14 line 5 up. "... panel, site 1)."

RESPONSE: We have added 'site #1' to make it clearer.

6. p15 line 4. "... panel, sites 2, 3)"

RESPONSE: We have added 'sites #2 and #3'.

7. p29 line 11. "complementary"

RESPONSE: This has been corrected.

8. Fig 1 B-D What is the unit on the y axis? Fraction, I assume, but what exactly?

RESPONSE: The unit on the y axis is the density of mutations or CPD lesions, which was calculated by determining the number of mutations or CPDs at each location relative to a TFBS midpoint (i.e., -100 to +100 nucleotides) and dividing this number by the number of nucleotide base-pairs analyzed at that location (equivalent to the number of TFBS). This is explained in the Methods section of the manuscript (see top of page 28).

9. p41 Fig 4 legend. The "lower panel" seems to now be Supplementary Tables S1 and S2.

RESPONSE: We have corrected the legend.

Finally, we wish to thank all of the reviewers for their helpful comments and suggestions. We believe that these changes have significantly improved the clarity of the manuscript.

Sincerely,

John Wyrick (on behalf of the authors)

References:

1. Hollenhorst, P.C., McIntosh, L.P. & Graves, B.J. Genomic and biochemical insights into the specificity of ETS transcription factors. *Annu Rev Biochem* **80**, 437-71 (2011).

2. Batchelor, A.H., Piper, D.E., de la Brousse, F.C., McKnight, S.L. & Wolberger, C. The structure of GABPalpha/beta: an ETS domain- ankyrin repeat heterodimer bound to DNA. *Science* **279**, 1037-41 (1998).
3. Zhang, T. et al. SDHD Promoter Mutations Ablate GABP Transcription Factor Binding in Melanoma. *Cancer Res* **77**, 1649-1661 (2017).
4. Friedberg, E.C. et al. *DNA repair and Mutagenesis*, xxvii, 1118 p. (ASM Press, Washington, D.C., 2006).
5. Pfeifer, G.P., You, Y.H. & Besaratinia, A. Mutations induced by ultraviolet light. *Mutat Res* **571**, 19-31 (2005).
6. Hu, J., Adebali, O., Adar, S. & Sancar, A. Dynamic maps of UV damage formation and repair for the human genome. *Proc Natl Acad Sci U S A* (2017).

Reviewers' Comments:

Reviewer #1:

None

Reviewer #2:

Remarks to the Author:

The authors have done an outstanding job of responding to the concerns raised by three reviewers. They have incorporated more explanations and clarified significant aspects of the work to allow broader readership and understanding. The manuscript has been improved by the process and this study should have a major impact on the field.

Reviewer #3:

Remarks to the Author:

The revisions satisfactorily address this reviewer's concerns.

Response to reviewers:

The reviewers indicated that the previous revisions had satisfactorily addressed their concerns and had improved the manuscript. No additional concerns were mentioned by the reviewers.